# A note on generalized hydrodynamics: inhomogeneous fields and other concepts

**Benjamin Doyon and Takato Yoshimura**

Department of Mathematics, King's College London, Strand WC2R 2LS, UK

## Abstract

Generalized hydrodynamics (GHD) was proposed recently as a formulation of hydrodynamics for integrable systems, taking into account infinitely-many conservation laws. In this note we further develop the theory in various directions. By extending GHD to all commuting flows of the integrable model, we provide a full description of how to take into account weakly varying force fields, temperature fields and other inhomogeneous external fields within GHD. We expect this can be used, for instance, to characterize the non-equilibrium dynamics of one-dimensional Bose gases in trap potentials. We further show how the equations of state at the core of GHD follow from the continuity relation for entropy, and we show how to recover Euler-like equations and discuss possible viscosity terms.



# 1 Introduction

The emergence of hydrodynamics in many-body extended systems is based on the phenomenon of local entropy maximization (often referred to as local thermodynamic equilibrium) [1–5]. This is the phenomenon according to which, at large times, the system decomposes into slowly varying local "fluid cells" where homogeneous Gibbs states exist. At leading order in a derivative expansion, the ensuing dynamics on the Gibbs potentials is completely fixed by the local conservation laws – this is often referred to as "pure hydrodynamics", as viscosity terms are absent. This is a powerful description, replacing the full many-body evolution, either quantum or classical, by differential equations for the few (or at least fewer) relevant local state parameters. It allows for the precise description of large-scale structures and the unearthing of exact results, and its universal applicability has been demonstrated in various situations and models [6–9]. In particular, it provides striking results in the context of quantum transport far from equilibrium [10–17] (see also the review [18]).

Recently [19], see also the related work [20], the hydrodynamic idea was extended to many-body integrable systems, where infinitely-many conservation laws are present. In this context, entropy maximization is conjectured to generate states in the infinite-dimensional variety of so-called generalized Gibbs ensembles[1] (GGEs) [22, 23], which therefore are used to characterize fluid cells. In [19], it was shown, in general diagonal-scattering integrable models of quantum field theory including the Lieb-Liniger and sinh-Gordon models, that the infinite system of conservation laws – for the infinite number of GGE potentials – can be recast into a system of hydrodynamic equations for quantities characterizing occupations and densities of quasi-particle states. In [20], the same equations were obtained in integrable quantum Heisenberg chains (the derivation making use of an additional assumption about the underlying dynamics). Interestingly, as will be studied in a coming work, these equations appear to give a universal description of quantum and classical quasi-particle elastic scattering; they widely generalize, for instance, hydrodynamic equations proven to emerge in the classical hard-rods model [5, 24]. In the same context, the effect of a localized defect on the non-equilibrium transport in quantum chains was also analyzed in [25, 26]. In fact even in free models, the hydrodynamic idea, as a semi-classical approximation, has found many applications [27–32].

The purpose of this letter is to extend this "generalized hydrodynamics" (GHD) theory further, within the quantum framework. We start by reviewing the main results of GHD in Section 2. In Section 3, we show that the GGE equations of state, at the core of GHD, are consequences of hydrodynamic entropy conservation. In Section 4 we show how to represent the dynamics associated to all conservation laws, not just the Hamiltonian. Using this, in Section 5 we derive GHD equations in the presence of external inhomogeneous fields, including force fields. Finally, in Section 6 we connect with aspects of ordinary fluid dynamics, including a derivation of Euler equations and a proposition for possible viscosity terms. We provide additional details in appendices, and especially in Appendix A we discuss how, in general free models, weak space-time variations of local densities and currents at large times guarantee the emergence of local GGEs, hence of GHD.

We emphasize that the force-field equations obtained can serve as a powerful tool in describing the late time non-equilibrium dynamics of a one-dimensional Bose gas confined in a weakly-varying trap potential. We also note that, recently, an alternative method to incorporate the inhomogeneity introduced by an external potential in a one-dimensional conformal field theory was proposed in [33].

---

[1] This has been widely studied in quantum models, but similar ideas can be used within classical dynamics [21].

## 2  Review of GHD

In this section we recall some of the basic concepts developed in [19] and [20], concentrating on the approach taken in the former, which puts emphasis on hydrodynamics ideas. The basic objects in the hydrodynamic theory of many-body extended systems are the local conserved densities $q_i(x,t)$ and local currents $j_i(x,t)$. These are quantum operators satisfying, under unitary dynamics, the continuity relations, or conservation laws,

$$\partial_t q_i(x,t) + \partial_x j_i(x,t) = 0, \tag{1}$$

as a consequence of the total charge $Q_i := \int dx\, q_i(x,t)$ being conserved $\partial_t Q_i = 0$. The set of such local conservation laws is a characteristics of the many-body system.

In integrable systems, this set is infinite, and the charges $Q_i$ relevant to the problem span the space of pseudolocal conserved charges [34]. In particular, entropy maximization of local subsystems under constraints of these conservation laws, as occurs under unitary dynamics, gives rise to GGEs, formally described by density matrices of the form[2] $\exp\left[-\sum_i \beta_i Q_i\right]$. It was in fact shown rigorously [35] that, in homogeneous systems, the existence of the long time limit implies that the stationary state is a GGE, where the completeness of the space of pseudolocal charges plays an important role. We will denote averages in such GGEs as $\langle \cdots \rangle_{\underline{\beta}}$ (with $\underline{\beta} = (\beta_i)_i$), and, for lightness of notation,

$$\mathsf{q}_i := \langle q_i \rangle_{\underline{\beta}}, \quad \mathsf{j}_i := \langle j_i \rangle_{\underline{\beta}}. \tag{2}$$

The problem of pure generalized hydrodynamics, as formulated in [19], is a direct generalization of usual pure hydrodynamics (without viscosity): it is the continuity problem applied to local cells where independent entropy maximization has occurred. That is, one assumes $\underline{\beta} = \underline{\beta}(x,t)$, and writes

$$\partial_t \mathsf{q}_i + \partial_x \mathsf{j}_i = 0, \tag{3}$$

where $\mathsf{q}_i = \langle q_i \rangle_{\underline{\beta}(x,t)}$ and $\mathsf{j}_i = \langle j_i \rangle_{\underline{\beta}(x,t)}$.

A convenient way of fixing the hydrodynamic problem for a given model is to provide the equations of state: relations connecting averages of currents to averages of densities. The thermodynamic Bethe ansatz (TBA) formulation of GGE averages offers a powerful way of obtaining these equations of state. In this formulation, the most natural objects are the quasi-particles. Quasi-particles are parametrized by their internal quantum numbers $a$ (parametrizing the spectrum of the model) and a continuous "rapidity" parameter $\theta$. In this letter we concentrate on Galilean and relativistically invariant models, wherefore $\theta$ will be identified with the velocity (Galilean) or the rapidity (relativistic). We will use the combined parameter

$$\boldsymbol{\theta} = (\theta, a). \tag{4}$$

The fundamental object that complete the full specification of the model is the differential scattering $\varphi(\boldsymbol{\theta}, \boldsymbol{\theta}')$, describing the scattering between particles. By relativistic or Galilean invariance, it depends on the rapidities or velocities only through their differences $\theta - \theta'$. In this paper we keep the discussion general and do not specify any particular model (any particular choice of particle spectrum and differential scattering phase), except when stated otherwise.

A conserved charge $Q_i$ is characterized, in terms of quasi-particles, by its one-particle eigenvalue $h_i(\boldsymbol{\theta})$. It will be convenient to consider the linear space of pseudolocal conserved charges as a function space spanned by the $h_i$s: we will denote $Q[h]$ the conserved charge (a linear

---

[2]More precisely [35], the conserved densities $q_i$ form a basis for the Hilbert space $\mathcal{H}$ with inner product generated by their susceptibilities, and a GGE state is given by a path in a variety whose tangent space is $\mathcal{H}$.

functional of $h$) associated to one-particle eigenvalue $h(\theta)$, and likewise $q[h]$ and $j[h]$ for the density and current. The density and current operators are also linear functionals of $h$.[3] Therefore, in any state (it does not even need to be homogeneous or stationary), the averages $\langle q[h]\rangle$ and $\langle j[h]\rangle$ are linear functionals of $h$, and we may consider the kernels $\rho_{\mathrm{p}}(\boldsymbol{\theta})$ (a "quasi-particle density") and $\rho_{\mathrm{c}}(\boldsymbol{\theta})$ (a "current spectral density")

$$\langle q[h]\rangle = \int \mathrm{d}\boldsymbol{\theta}\, \rho_{\mathrm{p}}(\boldsymbol{\theta})h(\boldsymbol{\theta}), \quad \langle j[h]\rangle = \int \mathrm{d}\boldsymbol{\theta}\, \rho_{\mathrm{c}}(\boldsymbol{\theta})h(\boldsymbol{\theta}) \tag{5}$$

where here and below $\int \mathrm{d}\boldsymbol{\theta} = \sum_a \int \mathrm{d}\theta$. These kernels are characteristics of the state. One may conveniently introduce the *effective velocity* $v^{\mathrm{eff}}(\boldsymbol{\theta})$ which relates them:

$$\rho_{\mathrm{c}}(\boldsymbol{\theta}) =: v^{\mathrm{eff}}(\boldsymbol{\theta})\rho_{\mathrm{p}}(\boldsymbol{\theta}). \tag{6}$$

The GGE equations of state, obtained from the TBA quasi-particle picture, which is the requirement of the existence of $\underline{\beta}$ such that both

$$\langle q[h]\rangle = \langle q[h]\rangle_{\underline{\beta}} =: \mathsf{q}[h] \quad \text{and} \quad \langle j[h]\rangle = \langle j[h]\rangle_{\underline{\beta}} =: \mathsf{j}[h] \quad \text{for all } h, \tag{7}$$

are the following integral relations for these kernels [19] (here and below prime symbols (′) represent rapidity derivatives $\partial/\partial\theta$):

$$\frac{\rho_{\mathrm{c}}(\boldsymbol{\theta})}{\rho_{\mathrm{p}}(\boldsymbol{\theta})} = \frac{E'(\boldsymbol{\theta}) + \int \mathrm{d}\boldsymbol{\alpha}\, \varphi(\boldsymbol{\theta}, \boldsymbol{\alpha})\rho_{\mathrm{c}}(\boldsymbol{\alpha})}{p'(\boldsymbol{\theta}) + \int \mathrm{d}\boldsymbol{\alpha}\, \varphi(\boldsymbol{\theta}, \boldsymbol{\alpha})\rho_{\mathrm{p}}(\boldsymbol{\alpha})}, \tag{8}$$

where $E(\boldsymbol{\theta})$ and $p(\boldsymbol{\theta})$ are the energy and momentum of a particle of type $a$ at velocity or rapidity $\theta$. These relations are independent of the state itself, they characterize the family of GGE states for a given model. In terms, instead, of the doublet $\rho_p(\boldsymbol{\theta})$ and $v^{\mathrm{eff}}(\boldsymbol{\theta})$, the GGE equations of state can be represented as [19]

$$v^{\mathrm{eff}}(\boldsymbol{\theta}) = v^{\mathrm{gr}}(\boldsymbol{\theta}) + \int \mathrm{d}\boldsymbol{\alpha}\, \frac{\varphi(\boldsymbol{\theta}, \boldsymbol{\alpha})\rho_{\mathrm{p}}(\boldsymbol{\alpha})}{p'(\boldsymbol{\theta})}(v^{\mathrm{eff}}(\boldsymbol{\alpha}) - v^{\mathrm{eff}}(\boldsymbol{\theta})) \tag{9}$$

with the group velocity $v^{\mathrm{gr}}(\boldsymbol{\theta}) := E'(\boldsymbol{\theta})/p'(\boldsymbol{\theta})$ (that is, (8) and (9) are equivalent under (6)). In this form, the equations of state of integrable systems are seen as equations specifying an effective velocity of quasi-particles, as a modification of the group velocity that depends on both the model and the state.

GGE equations of states mean that $\rho_p(\boldsymbol{\theta})$ completely determine the state, as both $\mathsf{q}[h]$ and $\mathsf{j}[h]$ may be evaluated once it is known. Hence the function $\rho_p(\boldsymbol{\theta})$ is a state variable. Other state variables exist. A particularly useful one is the occupation number $n(\boldsymbol{\theta})$ (taking values in $[0,1]$). Given $n(\boldsymbol{\theta})$, consider the symmetric bilinear form[4]

$$(h, g) := \int \frac{\mathrm{d}\boldsymbol{\theta}}{2\pi} h(\boldsymbol{\theta})\, n(\boldsymbol{\theta})\, g^{\mathrm{dr}}(\boldsymbol{\theta}) \tag{10}$$

---

[3]This is because every matrix element of $q[h]$ and $j[h]$ is. Indeed, let $h_i$ be a basis of the space of one-particle eigenvalues of conserved charges. Then $Q[\sum_i a_i h_i] = \sum_i a_i Q[h_i]$ by linearity. Since $Q[h_i](x) = \int \mathrm{d}x\, q[h_i](x)$, by locality we must also have $q[\sum_i a_i h_i] = \sum_i a_i q[h_i]$ (this is up to a total derivative of a local field, which can be set to zero by our choice of the density $q[h_i]$). Thus linearity holds at the operator level, and therefore $q(x) = \int \mathrm{d}\theta\, h(\theta)\hat{q}(x;\theta)$ for some density $\hat{q}(x;\theta)$. For the current, linearity then follows from the general relation between matrix elements of currents and densities (see e.g. appendix D of [19], equation D10).

[4]Although the fact that the bilinear form $(h, g)$ is symmetric is not completely apparent from this definition, it can be proven from it [36] (see also the short proof given in [19]).

where the dressing operation is defined by solving

$$h^{\mathrm{dr}}(\boldsymbol{\theta}) = h(\boldsymbol{\theta}) + \int \frac{\mathrm{d}\boldsymbol{\alpha}}{2\pi} \, \varphi(\boldsymbol{\theta}, \boldsymbol{\alpha}) n(\boldsymbol{\alpha}) h^{\mathrm{dr}}(\boldsymbol{\alpha}). \tag{11}$$

Charge densities and currents are expressed in terms of $n(\boldsymbol{\theta})$ as [19]

$$\mathsf{q}[h] = (p', h), \quad \mathsf{j}[h] = (E', h). \tag{12}$$

The nonlinear relation between the state variables $\rho_p(\boldsymbol{\theta})$ and $n(\boldsymbol{\theta})$ is $2\pi\rho_p(\boldsymbol{\theta}) = n(\boldsymbol{\theta})(p')^{\mathrm{dr}}(\boldsymbol{\theta})$, and the effective velocity takes the simple form

$$v^{\mathrm{eff}}(\boldsymbol{\theta}) = \frac{(E')^{\mathrm{dr}}(\boldsymbol{\theta})}{(p')^{\mathrm{dr}}(\boldsymbol{\theta})} \tag{13}$$

(see [19] for more details).

Finally, as a consequence of completeness of the set of functions $h(\boldsymbol{\theta})$, the GHD equations (3) can be expressed in various forms, using either state variables:

$$\partial_t \rho_p(\boldsymbol{\theta}) + \partial_x (v^{\mathrm{eff}}(\boldsymbol{\theta})\rho_p(\boldsymbol{\theta})) = 0 \tag{14}$$

$$\partial_t n(\boldsymbol{\theta}) + v^{\mathrm{eff}}(\boldsymbol{\theta})\partial_x n(\boldsymbol{\theta}) = 0. \tag{15}$$

The first form is immediate, and the second form can be derived from the first using the equations of state. The second form, involving occupation numbers, is particularly useful to solve initial-domain-wall problems (again see [19] for details).

Showing that GGE equations of state do indeed emerge in nontrivial integrable models is of course extremely difficult, and will not be discussed here. See however appendix A for a general discussion of how the GGE equations of state may emerge in free-particle models.

## 3 GGE equations of state from hydrodynamic entropy conservation

It was noted in [19, 20] that the density of available states $\rho_s(\boldsymbol{\theta}) = \rho_p(\boldsymbol{\theta})/n(\boldsymbol{\theta})$, which takes the form

$$2\pi\rho_s(\boldsymbol{\theta}) := p'(\boldsymbol{\theta}) + \int \mathrm{d}\boldsymbol{\alpha} \, \varphi(\boldsymbol{\theta}, \boldsymbol{\alpha})\rho_p(\boldsymbol{\alpha}) = (p')^{\mathrm{dr}}(\boldsymbol{\theta}), \tag{16}$$

and the density of holes, defined as $\rho_h(\boldsymbol{\theta}) := \rho_s(\boldsymbol{\theta}) - \rho_p(\boldsymbol{\theta})$, also satisfy the continuity equation (14) (that is, the equation holds with the replacements $\rho_p(\boldsymbol{\theta}) \mapsto \rho_s(\boldsymbol{\theta})$ and $\rho_p(\boldsymbol{\theta}) \mapsto \rho_h(\boldsymbol{\theta})$). Further, the fact that (14) holds for the densities of particles, states and holes with the same effective velocity implies that the Yang-Yang entropy density also follows the same continuity equation. The entropy density is

$$s(\boldsymbol{\theta}) := \rho_s(\boldsymbol{\theta}) \log \rho_s(\boldsymbol{\theta}) - \rho_p(\boldsymbol{\theta}) \log \rho_p(\boldsymbol{\theta}) - \rho_h(\boldsymbol{\theta}) \log \rho_h(\boldsymbol{\theta}). \tag{17}$$

Its integral $\int \mathrm{d}\boldsymbol{\theta} \, s(\boldsymbol{\theta})$ gives the specific entropy of the fluid cell at position $x, t$ (that is, the specific von Neumann entropy of the local GGE). It is found in [19] that

$$\partial_t s(\boldsymbol{\theta}) + \partial_x (v^{\mathrm{eff}}(\boldsymbol{\theta}) s(\boldsymbol{\theta})) = 0. \tag{18}$$

The statement (18) provides an interesting physical interpretation of GGE equations of states. Indeed, in ordinary pure hydrodynamics (with finitely-many conservation laws), any

function of the state identified as an entropy must obey a similar, natural conservation law; the exact form of the entropy is therefore related to the fluid equations of state.[5] One may then postulate that local conservation of entropy $s(\boldsymbol{\theta})$ is a basic principle, in some way equivalent to the GGE equations of state.

Assume the following.

(i) There exists a functional of $\rho_p$, denoted $v^{\mathrm{eff}}(\boldsymbol{\theta})$, with the following property. Define the entropy density $s(\boldsymbol{\theta})$, as a functional of $\rho_p$, by (17) and (16). If a space-time dependent $\rho_p(\boldsymbol{\theta})$ is nonzero ($\rho_p(\boldsymbol{\theta}) \neq 0$ for all $\boldsymbol{\theta}$) and satisfies the continuity equation (14), then $s(\boldsymbol{\theta})$ satisfies the continuity equation (18).

(ii) $v^{\mathrm{eff}}(\boldsymbol{\theta}) \to v^{\mathrm{gr}}(\boldsymbol{\theta})$ as $\rho_p(\boldsymbol{\theta}) \to 0$ (uniformly in $\boldsymbol{\theta}$).

Then we show that the GGE equations of state (9) hold.

The proof is as follows. With $\rho_h(\boldsymbol{\theta}) = \rho_s(\boldsymbol{\theta}) - \rho_p(\boldsymbol{\theta})$, combining (14) and (18) gives

$$\Big(\partial_t \rho_s(\boldsymbol{\theta}) + \partial_x(v^{\mathrm{eff}}(\boldsymbol{\theta})\rho_s(\boldsymbol{\theta}))\Big) \log \frac{\rho_s(\boldsymbol{\theta})}{\rho_h(\boldsymbol{\theta})} = 0. \tag{19}$$

Using $\rho_p(\boldsymbol{\theta}) \neq 0$, we have $\log \frac{\rho_s(\boldsymbol{\theta})}{\rho_h(\boldsymbol{\theta})} \neq 0$ hence $\rho_s(\boldsymbol{\theta})$ satisfies the same continuity equation with velocity $v^{\mathrm{eff}}(\boldsymbol{\theta})$. Let us replace, in the continuity equation for $\rho_s(\boldsymbol{\theta})$, the constitutive relation (16). We obtain

$$0 = p'(\boldsymbol{\theta})\partial_x v^{\mathrm{eff}}(\boldsymbol{\theta}) \int \mathrm{d}\boldsymbol{\alpha}\, \varphi(\boldsymbol{\theta}, \boldsymbol{\alpha})\Big(\partial_t \rho_p(\boldsymbol{\alpha}) + \partial_x(v^{\mathrm{eff}}(\boldsymbol{\theta})\rho_p(\boldsymbol{\alpha}))\Big). \tag{20}$$

Using the continuity equation for $\rho_p(\boldsymbol{\alpha})$, we then find

$$0 = \partial_x\Big[p'(\boldsymbol{\theta})v^{\mathrm{eff}}(\boldsymbol{\theta}) + \int \mathrm{d}\boldsymbol{\alpha}\, \varphi(\boldsymbol{\theta}, \boldsymbol{\alpha})(v^{\mathrm{eff}}(\boldsymbol{\theta}) - v^{\mathrm{eff}}(\boldsymbol{\alpha}))\rho_p(\boldsymbol{\alpha}))\Big]. \tag{21}$$

Therefore the expression in the square brackets on the right-hand side of (21) must be independent of $x$. Since this holds for any $x$-dependent $\rho_p(\boldsymbol{\alpha})$, it must be independent of it. Using the condition that the limit $\rho_p(\boldsymbol{\alpha}) \to 0$ of the effective velocity gives the group velocity, we finally find (9) as claimed.

In relation to the above result, it has recently been pointed out that entropy conservation can be seen as the conservation of an effective Noether current associated to a certain symmetry emerging at late times [38, 39]. It would be illuminating to understand if similar concepts can be applied to the specific fluid entropy $s(x, t) = \int \mathrm{d}\theta\, s(\boldsymbol{\theta})$ (note that if we integrate (18) over $\boldsymbol{\theta}$, we obtain a conservation law for the specific entropy $s(x, t)$). This might be the case, as entropy conservation is a dynamical symmetry, and emerges only when the GHD description becomes sensible. In the context of classical many-body systems, for models that follow trajectories consistent with quasi-static processes in thermodynamics, a symmetry whose conserved charge is the entropy was found recently in [39]. Applying this finding to the present situation might shed some light on the role of entropy in non-equilibrium dynamics.

## 4 Equations of states and GHD on commuting flows

In integrable systems, one may consider flows generated not only by the Hamiltonian, but also by any other conserved quantity $Q_k$; this will be useful when studying the effect of force fields

---

[5]The entropy is also related to viscosity terms, which must account for positive entropy production.

in the next section. The goal of this section is to report on the main equations that generalize GHD to such commuting flows. Since the conserved charges $Q_k$ are linear functionals of the one-particle eigenvalues $h_k$, we will also use the notation $Q_k = Q[h_k]$.

Let us denote by $t_k$ the associated "time", $\partial_{t_k} \mathcal{O} := i[Q_k, \mathcal{O}]$ (with $t_1 = t$ the ordinary time, under Hamiltonian evolution $Q_1 = H$). By involution, all flows commute, wherefore conserved quantities $Q_i$ are also conserved with respect to all $t_k$ evolutions. There are associated currents $j_{k,i}$:

$$\partial_{t_k} q_i + \partial_x j_{k,i} = 0, \tag{22}$$

which are bilinear functionals of $h_k$ and $h_i$, denoted by $j_{k,i} = j[h_k, h_i]$ (we will also use the notations $\mathrm{j}_{k,i}$ and $\mathrm{j}[h_k, h_i]$ for averages in GGEs)[6]. Generalized hydrodynamics may also be applied to all these flows. Below we assume that the set of local conserved charges in involution, from which GGEs are formed, with respect to time $t_k$, is the same as that with respect to the original time $t$ – this is usually case in integrable systems. Thus local entropy maximization is described by the same set of GGE states. By commutativity of the flows, under local entropy maximization, local GGE potentials are well-defined functions simultaneously of all time variables, $\underline{\beta} = \underline{\beta}(x, \{t\})$, and we have

$$\partial_{t_k} \mathrm{q}_i + \partial_x \mathrm{j}_{k,i} = 0. \tag{23}$$

Note that the currents $j_{k,i}$ are fixed by conservation, (22), only up to the addition of a constant times the identity operator. We fix this gauge freedom, implicitly, by providing explicit expressions for these currents in GGE states below.

Bilinearity implies, in general states, the existence of the kernel $\rho_c(\gamma, \theta)$ (by abuse of notation, we use the same symbol $\rho_c$ as in (5) but with two rapidity arguments in order to represent this new kernel) such that

$$\langle j[h, g] \rangle = \int d\gamma d\theta \, \rho_c(\gamma, \theta) h(\gamma) g(\theta). \tag{24}$$

The GGE equations of state encompass relations between this kernel and $\rho_p(\theta)$, generalizing (8) in a natural manner. This can be obtained following the derivation of [19] and using the results of [19, App D]:

$$\frac{\rho_c(\gamma, \theta)}{\rho_p(\theta)} = \frac{\partial_\theta \delta(\theta - \gamma) + \int d\alpha \, \varphi(\theta, \alpha) \rho_c(\gamma, \alpha)}{p'(\theta) + \int d\alpha \, \varphi(\theta, \alpha) \rho_p(\alpha)}. \tag{25}$$

This is the most general form of the equations of state, as integrating over $\gamma$ against $E(\gamma)$ reproduces the GGE equations of state for the usual time evolution. Likewise, one may define a $\gamma$-dependent group velocity $v^{gr}(\gamma, \theta) := \partial_\theta \delta(\theta - \gamma)/p'(\theta)$ and a $\gamma$-dependent effective velocity

$$\rho_c(\gamma, \theta) =: v^{eff}(\gamma, \theta) \rho_p(\theta), \tag{26}$$

and the equations of state (25) are equivalent to

$$v^{eff}(\gamma, \theta) = v^{gr}(\gamma, \theta) + \int d\alpha \, \frac{\varphi(\theta, \alpha) \rho_p(\alpha)}{p'(\theta)} (v^{eff}(\gamma, \alpha) - v^{eff}(\gamma, \theta)). \tag{27}$$

Using the bilinear form (10), results of [19, App D] also enable us to express the density and current associated to a conserved charge $Q_k$, in any GGE state parametrized by the occupation

---

[6]Linearity of $j_{k,i}$ as a functional of $h_k$ follows from (22) and the fact that $\partial_{t_k} q_i = i[Q[h_k], q_i]$.

number $n(\boldsymbol{\theta})$, as follows:[7]

$$\mathbf{q}[h] = \big(p', h\big), \quad \mathbf{j}[h, g] = (h', g). \tag{28}$$

Note that integrating $\rho_{\mathrm{c}}(\boldsymbol{\gamma}, \boldsymbol{\theta})$ (resp. $v^{\mathrm{eff}}(\boldsymbol{\gamma}, \boldsymbol{\theta})$) against $h(\boldsymbol{\gamma})$ gives a current spectral density $\rho_{\mathrm{c}}[h](\boldsymbol{\theta})$ (resp. the effective velocity $v^{\mathrm{eff}}[h](\boldsymbol{\theta})$) corresponding to a flow produced by $Q[h]$, and we have

$$2\pi \int \mathrm{d}\boldsymbol{\gamma} \, h(\boldsymbol{\gamma}) \rho_{\mathrm{c}}(\boldsymbol{\gamma}, \boldsymbol{\theta}) =: 2\pi \rho_{\mathrm{c}}[h](\boldsymbol{\theta}) = n(\boldsymbol{\theta})(h')^{\mathrm{dr}}(\boldsymbol{\theta}) \tag{29}$$

and

$$\int \mathrm{d}\boldsymbol{\gamma} \, h(\boldsymbol{\gamma}) v^{\mathrm{eff}}(\boldsymbol{\gamma}, \boldsymbol{\theta}) =: v^{\mathrm{eff}}[h](\boldsymbol{\theta}) = \frac{(h')^{\mathrm{dr}}(\boldsymbol{\theta})}{(p')^{\mathrm{dr}}(\boldsymbol{\theta})}. \tag{30}$$

with the usual effective velocity being $v^{\mathrm{eff}}(\boldsymbol{\theta}) = v^{\mathrm{eff}}[E](\boldsymbol{\theta})$. We have the equations of state

$$\frac{\rho_{\mathrm{c}}[h](\boldsymbol{\theta})}{\rho_{\mathrm{p}}(\boldsymbol{\theta})} = \frac{h'(\boldsymbol{\theta}) + \int \mathrm{d}\boldsymbol{\alpha} \, \varphi(\boldsymbol{\theta}, \boldsymbol{\alpha}) \rho_{\mathrm{c}}[h](\boldsymbol{\alpha})}{p'(\boldsymbol{\theta}) + \int \mathrm{d}\boldsymbol{\alpha} \, \varphi(\boldsymbol{\theta}, \boldsymbol{\alpha}) \rho_{\mathrm{p}}(\boldsymbol{\alpha})}. \tag{31}$$

or equivalently

$$v^{\mathrm{eff}}[h](\boldsymbol{\theta}) = \frac{h'(\boldsymbol{\theta})}{p'(\boldsymbol{\theta})} + \int \mathrm{d}\boldsymbol{\alpha} \, \frac{\varphi(\boldsymbol{\theta}, \boldsymbol{\alpha}) \rho_{\mathrm{p}}(\boldsymbol{\alpha})}{p'(\boldsymbol{\theta})} (v^{\mathrm{eff}}[h](\boldsymbol{\alpha}) - v^{\mathrm{eff}}[h](\boldsymbol{\theta})). \tag{32}$$

The generalized hydrodynamic problem (23) including all commuting flows of a given integrable model can then be recast as follows. Consider times $t^h$ generated by $Q[h]$ (that is, $\partial_{t^h} \mathcal{O} = \mathrm{i}[Q[h], \mathcal{O}]$). Then

$$\partial_{t^h} \rho_{\mathrm{p}}(\boldsymbol{\theta}) + \partial_x \big(v^{\mathrm{eff}}[h](\boldsymbol{\theta}) \rho_{\mathrm{p}}(\boldsymbol{\theta})\big) = 0 \tag{33}$$

with equations of state (30). Following the derivation of [19], the GHD equations for arbitrary flows can also be written in terms of occupation number variables $n(\boldsymbol{\theta})$,

$$\partial_{t^h} n(\boldsymbol{\theta}) + v^{\mathrm{eff}}[h](\boldsymbol{\theta}) \partial_x n(\boldsymbol{\theta}) = 0. \tag{34}$$

Finally, commuting-flow continuity equations hold for state and hole densities, as well as for the density of the entropy:

$$\partial_{t^h} s(\boldsymbol{\theta}) + \partial_x \big(v^{\mathrm{eff}}[h](\boldsymbol{\theta}) s(\boldsymbol{\theta})\big) = 0. \tag{35}$$

## 5 Evolution in inhomogeneous fields

It is natural and physically meaningful to consider how external potentials, temperature fields, or inhomogeneous fields associated to other conserved quantities modify the GHD equations (14), (15).

Recall the basic hydrodynamic assumption that averages of local densities and currents, in an inhomogeneous state, may be approximated by averages in a homogeneous, entropy-maximized state, with inhomogeneous potentials. This approximation leads to Euler-type hydrodynamic equations. These equations are first-order differential equations for hydrodynamic

---

[7]The second of Equations (28) was explicitly obtained in [19, App D] (see Eq. (D13)), but only for $h(\theta)$ with certain properties – corresponding, in the sinh-Gordon model, to time evolution with respect to local charges. It is natural, however, to assume that the same form holds for any quasi-local charge, and it is under this assumption that (28) is written in this general form.

variables, which may be taken as the densities, or as the inhomogeneous potentials themselves. The hydrodynamic assumption is expected to be a good approximation when variations of densities and currents occur on large scales, so that locally, the state looks homogeneous. The Euler-type hydrodynamic equations are in fact the leading terms in a derivative expansion; higher derivative terms would give rise to viscosity and other effects. It is within this picture that we may consider external inhomogeneous fields affecting the time evolution. We assume that these external fields also only display variations on large scales, so that locally the evolution looks homogeneous. The Euler-type hydrodynamic equations obtained will therefore again be leading-order terms in a derivative expansions, neglecting any term containing derivatives of hydrodynamic variables or potentials with a total order of 2 or more. The equations are simply obtained by deriving leading-order evolution equations at the operator level, and then using the hydrodynamic approximation of local entropy maximization.

Inhomogeneous fields, of course, break the integrability of the dynamics. However, since locally the dynamics still looks like a homogeneous evolution with respect to an integrable Hamiltonian, the local GGE approximation stays valid under time evolution. This is made clear below, as we show that the first-derivative-order approximation of the dynamics leads to a consistent equation for local GGE states (the initial state does not know about the dynamics, and thus can be chosen within the space of local GGE states). This is certainly not surprising. In usual hydrodynamics (such as for water waves), local fluid cells are approximated by Galilean boosts of equilibrium states, thus involve the momentum operator. In an inhomogeneous field, this description still holds and Euler and Navier-Stokes equations with force terms give a good description. This is so even though inhomogeneity breaks translation invariance (thus the momentum operator is not a conserved quantity of the dynamics). The derivation below is simply a generalization of this fact to infinitely-many conserved charges. Naturally, the higher-order derivatives terms neglected would modify this picture, and may be expected to lead to integrability breaking effects at large times. But this is beyond the scope of this work.

To start with, let us briefly recall a typical case in relativistic one-dimensional quantum field theory with $U(1)$ symmetry: coupling the particle current $J^\mu(x,t)$ to an external electric field, $A^\mu(x) = (V_0(x), 0)$ where $V_0(x)$ is the electric potential[8]. Here in order to fix the notation, we assume the particle current is associated to some conserved charge $Q_0$ (that is, $J^0(x,t) = q_0(x,t)$ and $J^1(x,t) = j_0(x,t)$)), and we take $Q_1 = H$ to be the total energy without external field. The external field deforms the evolution Hamiltonian in a familiar fashion:

$$H_{\text{force}} = H - \int dx\, A_\mu(x) J^\mu(x,t) \tag{36}$$

$$= H + \int dx\, V_0(x) q_0(x,t). \tag{37}$$

Accordingly the hydrodynamic conservation equations become [42] (keeping the $(x,t)$-dependence implicit), at the first order in a derivative expansion,

$$\partial_\nu \langle T^{\mu\nu} \rangle = F^{\mu\nu} \langle J_\nu \rangle, \quad \partial_\mu \langle J^\mu \rangle = 0 \tag{38}$$

where $T^{\mu\nu}$ is the energy-momentum tensor, $F^{01} = -F^{10} = \partial_x V_0$ and $F^{00} = F^{11} = 0$, and averages are taken in local fluid cells. Alternatively this can be written as

$$\partial_t \mathsf{q}_1 + \partial_x \mathsf{j}_1 + (\partial_x V_0) \mathsf{j}_0 = 0, \quad \partial_t \mathsf{q}_0 + \partial_x \mathsf{j}_0 = 0. \tag{39}$$

We now generalize this, as well as more complicated external fields, to GHD.

In order to have a clearer general framework, we divide the external field, in general, into two types. We first understand an external force field, arising from a potential $V_0(x)$, as a

---

[8]We choose the metric $\eta^{\mu\nu} = \text{diag}(-1,1)$.

field coupled to a conserved density $q_0(x) = q[h_0](x)$ which has the property the associated conserved charge $Q_0 = \int dx\, q_0(x)$ commutes with all conserved densities $q_i(x)$:

$$[Q_0, q_i] = 0. \tag{40}$$

This is a sensible definition of an external potential $V_0$, as it implies that physical quantities in GGEs only depend on potential differences. Indeed, if $V_0(x) = V_0$ is independent of $x$, then $\int dx\, V_0(x)q_0(x) = V_0 Q_0$, and as a consequence of (40), evolution of local densities with respect to $H + V_0 Q_0$, and averages of local densities with respect to density matrices of the form $e^{-\sum_i \beta_i Q_i - V_0 Q_0}$, are independent of $V_0$. Note that thanks to (40), all currents associated to the $Q_0$ evolution must vanish[9], $j_{0,i} = 0$. Therefore, using (28), the one-particle eigenvalue $h_0(\boldsymbol{\theta})$ must be independent of the rapidity, $h_0(\theta, a) = h_0(a)$ (that is, $h_0'(\boldsymbol{\theta}) = 0$). As an example, in the Lieb-Liniger model (a Galilean model with one particle type only) one may choose $Q_0$ to be the number operator, which counts the number of quasi-particles, $h_0(\theta) = 1$. In a model with an internal charge $a \in \{+1, -1\}$, such as the (relativistic) sine-Gordon mode, one may take $Q_0$ to be the total charge, with $h_0(\theta, a) = a$.

We are thus interested, in a first instance, in deriving a force-field, pure hydrodynamic equation describing the time derivative of local conserved densities under the time evolution with respect to the force-field Hamiltonian,

$$\partial_t \mathcal{O} = i[H_{\text{force}}, \mathcal{O}], \quad H_{\text{force}} = H + \int dx\, V_0(x)q_0(x). \tag{41}$$

We show in Appendix B that the infinite set of force-field hydrodynamic equations, under the assumption both of local entropy maximization and of weak spacial variations of the potential $V_0(x)$, are

$$\partial_t q_i + \partial_x j_i + (\partial_x V_0)j_{i,0} = 0 \quad \text{(force field)}. \tag{42}$$

We see that the force term, proportional to the space derivative $\partial_x V_0$ of the potential, involves the charge current associated to the time evolution with respect to $Q_i$ (see (23)). Specializing to the energy $Q_1 = H$ (choosing $i = 1$), we observe that the force term controlling the continuity equation for the energy density is proportional to the usual particle current $j_{1,0} = j_0$, as is intuitively clear and in agreement with (39). Equation (42) is to be seen as the leading part of a derivative expansion, where neglected terms are higher space derivatives in the potential and in conserved densities and currents.

In a second instance, we consider more general external fields, associated to general conserved densities. These are perturbations of the type $\int dx \sum_k V_k(x)q_k(x)$:

$$\partial_t \mathcal{O} = i[H_{\text{field}}, \mathcal{O}], \quad H_{\text{field}} = H + \sum_k \int dx\, V_k(x)q_k(x). \tag{43}$$

For instance, as $q_1(x)$ is the energy density (according to the convention we use), the term $\int dx\, V_1(x)q_1(x)$ may be understood as a perturbation by an inhomogeneous temperature field, with $x$-dependent temperature $(V_1(x))^{-1}$ (this interpretation being valid under the hydrodynamic assumption, with weak variations). It is useful to introduce the one-particle potential

$$W(x) := \sum_k V_k(x)h_k, \tag{44}$$

---

[9]More precisely the argument is as follows. The current must satisfy $\partial_x j_{0,i}(x) = 0$. In QFT, this implies that $j_{0,i}(x)$ is proportional to the identity operator. Hence its GGE average is independent of the potentials $\underline{\beta}$, hence independent of $n(\boldsymbol{\theta})$. Using (28), we find that $h_0' = 0$, and thus the constant must be zero.

which is the one-particle eigenvalue function of the operator $\sum_k V_k(x)Q_k$ (in this notation, $W(x)$ is, implicitly, a function of $\boldsymbol{\theta}$). Using $q_k(x) = q[h_k](x)$, the perturbation is written in a somewhat more general way in terms of any $W(x)$:

$$H_{\text{field}} = H + \int \mathrm{d}x \, q[W(x)](x). \tag{45}$$

We show in Appendix B that the infinite set of hydrodynamic equations in inhomogeneous fields, again under the assumption both of local entropy maximization and of weak spacial variations of the potentials $V_k(x)$ (weak spacial variations of $W(x)$), are

$$\partial_t \mathsf{q}_i + \partial_x \mathsf{j}_i + \sum_k \left( \partial_x (V_k \mathsf{j}_{k,i}) + (\partial_x V_k) \, \mathsf{j}_{i,k} \right) = 0 \tag{46}$$

or equivalently

$$\partial_t \mathsf{q}_i + \partial_x \left( \mathsf{j}_i + \mathsf{j}[W, h_i] \right) + \mathsf{j}[h_i, \partial_x W] = 0. \tag{47}$$

These generalize (42), which is recovered by choosing $V_k(x) = 0$ for all $k \geq 1$ and using $\mathsf{j}_{0,i} = 0$.

Equations (42), (46) and (47) are derived without invoking integrability, which is only included in the fact that there are infinitely-many of these equations. They are valid under the following assumptions. First, there is the usual hydrodynamic assumption that local averages of densities and currents are well approximated by averages within local GGEs. Second, all higher-derivative terms occurring from the quantum dynamics with respect to $H_{\text{field}}$ are neglected. These are terms composed of products of the first or higher derivative of the potentials $V_k(x)$, times local fields and their derivatives, with, in total, two or more space derivatives. As long as the potentials are varying in a smooth enough fashion, such higher-derivative terms are indeed negligible. We recall that the assumption of local GGEs (which comes from that of local entropy maximization) gives rise to a continuity equation, which is a first-derivative equation. It therefore only gives the leading first-derivative terms in a derivative expansion of the full hydrodynamics (thus neglects higher derivatives of hydrodynamic variables such as viscosity terms). Assuming that variations of the potentials are of the order of the variations of the hydrodynamic variables, it is thus consistent to neglect higher derivative terms as above, and to keep the total number of derivatives, of hydrodynamic variables and potentials, to a maximum of 1. Of course, such derivative expansions, common in hydrodynamic problems, are not *controlled* approximations, and it is difficult to evaluate the corrections.

Further, we show in Appendix B that (46), (47) can be recast, in the quasi-particle basis, into the following equivalent equations for the occupation number $n(\theta)$ and for the densities (here keep implicit the $x$ and $t$ dependencies),

$$\partial_t n(\boldsymbol{\theta}) + v^{\text{eff}}[E+W](\boldsymbol{\theta})\partial_x n(\boldsymbol{\theta}) + a^{\text{eff}}(\boldsymbol{\theta})\partial_\theta n(\boldsymbol{\theta}) = 0 \tag{48}$$

and

$$\partial_t \rho(\boldsymbol{\theta}) + \partial_x \left( v^{\text{eff}}[E+W](\boldsymbol{\theta})\rho(\boldsymbol{\theta}) \right) + \partial_\theta \left( a^{\text{eff}}(\boldsymbol{\theta})\rho(\boldsymbol{\theta}) \right) = 0, \tag{49}$$

which holds for $\rho = \rho_{\text{s}}$, $\rho_{\text{p}}$ and $\rho_{\text{h}}$. Recall that $E$ is the function of $\boldsymbol{\theta}$ giving the one-particle energy (the Hamiltonian one-particle eigenvalue). Here the effective acceleration is

$$a^{\text{eff}}(\boldsymbol{\theta}) = \frac{F^{\text{dr}}(\boldsymbol{\theta})}{(p')^{\text{dr}}(\boldsymbol{\theta})}. \tag{50}$$

The space-dependent force function $F(\boldsymbol{\theta})$ is the derivative of the total energy $E(\boldsymbol{\theta}) + W(\boldsymbol{\theta})$ with respect to space; since $E(\boldsymbol{\theta})$ is independent of space, this is

$$F(\boldsymbol{\theta}) = -\sum_k h_k(\boldsymbol{\theta})\partial_x V_k = -\partial_x W(\boldsymbol{\theta}). \tag{51}$$

The effective velocity $v^{\text{eff}}[E + W](\boldsymbol{\theta})$ depends on $x$ both through the one-particle potential $W(\boldsymbol{\theta}) = W(x; \boldsymbol{\theta})$, which modifies the local energy to $E(\boldsymbol{\theta}) + W(x; \boldsymbol{\theta})$, and thus modifies the local group velocity; and through the $(x, t)$-dependent occupation number $n(\boldsymbol{\theta})$, or particle density $\rho_{\text{p}}(\boldsymbol{\theta})$, which determines it (see (30) and (32)). Likewise, the effective acceleration depends on $x$ both through the potentials and through the dressing operation.

Equations (48) and (49) invoke integrability in the use of the TBA formalism, and of the completeness of the space of pseudolocal conserved charges. They are otherwise both direct consequences of (46) (or equivalently (47)), without further approximation. They use the quasi-particle expressions of the local GGE densities and currents that are involved in (46), (47).

We see that the effects of the potential $W(x)$ (or equivalently $V_k(x)$) are twofold. First, there is a modification of the effective velocity to $v^{\text{eff}}(\boldsymbol{\theta}) = v^{\text{eff}}[E](\boldsymbol{\theta}) \mapsto v^{\text{eff}}[E+W](\boldsymbol{\theta})$, which takes into account the *local potential* $W$ at the position $x$. Second, there is an extra term involving $\theta$ derivatives, which takes into account the acceleration due to *spacial variations* of $W$ around the position $x$. We note that since $v^{\text{eff}}[E + W](\boldsymbol{\theta})$ only involves $\theta$-derivatives $W(x)'$ of the one-particle potential, and since $h_0' = 0$, it is clear that the force-field potential $V_0(x)$ does not affect the effective velocity. A force field only leads to an acceleration, without modifying the local effective velocity. Other external fields such as temperature fields, however, do modify the local effective velocity.

Consider a pure force field in a Galilean model with a single-particle spectrum (such as the Lieb-Liniger model). In this case, we have $\boldsymbol{\theta} = \theta$, $h_0(\theta) = 1$ and $p(\theta) = m\theta$. Then, the effective acceleration $a^{\text{eff}}(\theta)$ simplifies to the usual acceleration, independently of $\theta$,

$$a^{\text{eff}}(\theta) = -\partial_x V_0 / m \quad \text{(Galilean, single-particle spectrum, pure force field).} \qquad (52)$$

Equation (48) (or equivalently (49)) represents evolution in the presence of space-dependent external fields; it is valid in the limit of weak variations of both the hydrodynamic quantities and of the potentials themselves. As it is a pure-hydrodynamic equation, it does not take into account any viscosity effects, which give rise to terms with higher derivatives of the hydrodynamic variables, or, similarly, any effect related to the presence of nonzero higher derivatives of the potentials. In a pure force field, $V_{k \geq 1} = 0$, the effective velocity is not affected, and if the force field is constant, $\partial_x^2 V_0(x) = 0$, the effective acceleration does not depend on space. In this case, one may argue that, as usual, at large times variations of hydrodynamic variables become smaller, and thus pure hydrodynamics provides a good description.[10] Otherwise, spacial variations of potentials are present in the pure hydrodynamic equations, and as they do not change with times, they will fix a minimum spacial-variation scale for the hydrodynamic variables. Thus, in this case, the pure hydrodynamic equations cannot become more accurate at large times, and we must understand (48) as being valid for *a finite period of time*, whose extent depends on the size of spacial variations of the potentials. During this time, discrepancies between the predictions of (48) and the actual evolution, due to neglected terms whose amplitude does not decay, accumulate. Beyond this time, one might expect the integrability-breaking effects of the presence of space-varying potentials to become important.

Let us now investigate stationary solutions to the force-field equations (48). Consider a fluid state which is, at every position $x$, the Gibbs state associated to $H + \sum_k V_k(x) Q_k$ at the temperature $\beta^{-1}$ (independent of $x$). This is the local density approximation of the finite-temperature, inhomogeneous state $e^{-\beta H_{\text{field}}}$. We show that the one-parameter family of such local-Gibbs states, parametrized by the temperature $\beta^{-1}$, is indeed a stationary solution to (48).

---

[10] In fact, in this case, if the force is nonzero, one has to consider carefully the large-distance asymptotics of hydrodynamic variables, a subject which is beyond the scope of this paper.

For this purpose, consider the one-particle eigenvalue $w(\boldsymbol{\theta}) = \sum_i \beta_i h_i(\boldsymbol{\theta})$ of the operator in the exponent in the GGE density matrix $\exp[-\sum_i \beta_i Q_i]$. The function $w(\boldsymbol{\theta})$ is yet another GGE state variable. For instance, by standard (G)TBA arguments [36,37], it is related to the occupation number $n(\boldsymbol{\theta})$ as follows: setting the pseudoenergy to be

$$\epsilon(\boldsymbol{\theta}) = \log(1 - n(\boldsymbol{\theta})) - \log(n(\boldsymbol{\theta})), \tag{53}$$

we have

$$w(\boldsymbol{\theta}) = \epsilon(\boldsymbol{\theta}) + \int \frac{\mathrm{d}\boldsymbol{\alpha}}{2\pi} \varphi(\boldsymbol{\theta}, \boldsymbol{\alpha}) \log(1 + e^{-\epsilon(\boldsymbol{\alpha})}). \tag{54}$$

Clearly $\epsilon(\boldsymbol{\theta})$ satisfies the same equation (48) as does $n(\boldsymbol{\theta})$. Note that $\partial_x \epsilon(\boldsymbol{\theta}) = (\partial_x w)^{\mathrm{dr}}(\boldsymbol{\theta})$, and that, using the fact that $\varphi(\boldsymbol{\theta}, \boldsymbol{\alpha})$ depends on the rapidities through their difference $\theta - \alpha$ only, $\partial_\theta \epsilon(\boldsymbol{\theta}) = (\partial_\theta w)^{\mathrm{dr}}(\boldsymbol{\theta})$ (we recall that the superscript $^{\mathrm{dr}}$ indicates dressed quantities as per (11)). Using these statements and setting $\partial_t n(\boldsymbol{\theta}) = 0$, one finds that in terms of the local-GGE one-particle eigenvalue $w(\boldsymbol{\theta})$, a stationary solution satisfies the equation

$$\frac{(\partial_x w)^{\mathrm{dr}}(\boldsymbol{\theta})}{(\partial_x W)^{\mathrm{dr}}(\boldsymbol{\theta})} = \frac{(\partial_\theta w)^{\mathrm{dr}}(\boldsymbol{\theta})}{(\partial_\theta(E + W))^{\mathrm{dr}}(\boldsymbol{\theta})} \tag{55}$$

(we also used (50), (16) and (13)). It is simple to see that

$$w = \beta (E + W) \tag{56}$$

is a solution to this equation for any $\beta$ (recall that $E = E(\boldsymbol{\theta})$ depends on $\boldsymbol{\theta} = (\theta, a)$ but not on $x$, and that $W = W(x) = W(x)(\boldsymbol{\theta})$ depends on both $x$ and $\boldsymbol{\theta}$). This is the local density approximation of the state $e^{-\beta H_{\mathrm{field}}}$.

The above statement is very natural physically. Assuming that the spatial variations of the potential occur only on large distance scales, we do not expect these inhomogeneities to lead to localization (the latter would of course break the hydrodynamic assumptions). Yet, we expect inhomogeneities to lead to integrability-breaking effects. Therefore, at very large times, after integrability-breaking effects have arisen, we expect the stationary-state density matrix to be of the thermal form $e^{-\beta H_{\mathrm{field}}}$ for some $\beta$. In such a state, variations of all densities and currents occur on large distance scales, hence this is well approximated by a local density approximation – the "fluid form" of this state. Further, since the force-field hydrodynamic equations should approximate well the dynamics when variations are on large scales, we expect this approximation to be a stationary solution to these equations, as indeed shown above. That is, although the force-field equations do not contain all integrability breaking effects and might not by themselves lead to thermalization, the thermalized state should be stationary with respect to it. This is much like the fact that the ideal-gas distribution is invariant under the free-particle evolution, although it may only arise, physically, as a consequence of the small interactions between the particles of the gas.

We have not established uniqueness of this stationary solution to the force-field equations – in particular, it is simple to see that in the case of free-particle models, any function $f(E + W)$ is a solution. One may wonder if, similarly, there are additional stationary solutions in interacting integrable models, and if these make physical sense. One may also wonder what, if any, stationary solution is actually reached at long times from solving the pure hydrodynamic equations (48) without higher-derivative terms. If it is not the local-Gibbs state above, then this might correspond to a "pre-thermalization" plateau, which appears before the integrability-breaking effects of the inhomogeneous potential become important. We leave these questions for future works.

# 6 Euler and Navier-Stokes equations

An important ingredient in conventional hydrodynamics is what is often referred to as the Euler equation: this is a continuity equation relating the fluid velocity $v$ to the internal pressure $\mathscr{P}$ and the fluid's mass density $\rho_{\mathrm{fl}}$:

$$\partial_t v + v \partial_x v = -\frac{1}{\rho_{\mathrm{fl}}} \partial_x \mathscr{P}. \tag{57}$$

It is a simple consequence of conservation of the mass density and mass current, and expresses the variation of the fluid's velocity as a convection term and a term due to pressure variations.

In generalized hydrodynamics, such equations also arise in a natural fashion. It is obvious from the symmetry of the bilinear form (10) that, in any GGE state, the current associated to the conserved quantity with one-particle eigenvalue $h(\boldsymbol{\theta}) = p'(\boldsymbol{\theta})$ is equal to the density associated to $h(\boldsymbol{\theta}) = E'(\boldsymbol{\theta})$:

$$\mathsf{j}[p'] = \mathsf{q}[E']. \tag{58}$$

For instance, in Galilean invariant systems, $p'(\boldsymbol{\theta}) = m_a$ is the mass of the particle, and its momentum is $E'(\boldsymbol{\theta}) = p(\boldsymbol{\theta})$, and this is equality between mass current and momentum density. In relativistic system, $p'(\boldsymbol{\theta}) = E(\boldsymbol{\theta})$ and $E'(\boldsymbol{\theta}) = p(\boldsymbol{\theta})$, so this is instead equality between energy current and momentum density (which amounts to the fact that the energy-momentum tensor is symmetric).

Let us then define the fluid velocity $v$ as follows:

$$\mathsf{j}[p'] =: v \, \mathsf{q}[p']. \tag{59}$$

This is the velocity for the mass current (Galilean) or energy current (relativistic). The quantity $v$ depends on $x$ and $t$ (but is of course independent of $\boldsymbol{\theta}$). The two conservation laws $\partial_t \mathsf{q}[p'] + \partial_x \mathsf{j}[p'] = 0$ and $\partial_t \mathsf{q}[E'] + \partial_x \mathsf{j}[E'] = 0$ then immediately imply

$$\mathsf{q}[p'] \partial_t v + \partial_x \mathsf{j}[E'] - v \, \partial_x (v \, \mathsf{q}[p']) = 0. \tag{60}$$

We may then define the fluid mass density and pressure as

$$\rho_{\mathrm{fl}} := \mathsf{q}[p'], \quad \mathscr{P} := \mathsf{j}[p] - \rho_{\mathrm{fl}} v^2, \tag{61}$$

and we recover (57), using $E'(\boldsymbol{\theta}) = p(\boldsymbol{\theta})$. The interpretation of the above identification is particularly clear with Galilean invariance. In this case $\rho_{\mathrm{fl}}$ is indeed the physical mass density, and the second equation in (61) is the correct relation between the momentum current $\mathsf{j}[p]$ and the pressure $\mathscr{P}$: it identifies the momentum current as the internal pressure plus $v$ times the current associated to the displacement of the fluid cell $\rho_{\mathrm{fl}} v$. In the relativistic case, $\rho_{\mathrm{fl}}$ is the energy density, and $\mathscr{P}$ has a similar interpretation.

Notice that the physical pressure $\mathscr{P}$ is *not* equal to the generalized specific free energy (free energy per unit volume) $f = \int \mathrm{d}p(\boldsymbol{\theta})/(2\pi) \log(1 + e^{-\epsilon(\boldsymbol{\theta})})$ (where the pseudoenergy is defined in (53)); this is of course natural in states that are not thermal Gibbs states. As such, unlike the case in conventional hydrodynamics, in the Galilean case the continuity equation for the energy $\partial_t \mathsf{q}[E] + \partial_x \mathsf{j}[E] = 0$ is no longer expressible in terms of the fluid velocity and thermodynamic variables.

It is also straightforward to generalize the above to the forced equation (42). Repeating the above derivation with the conservation laws $\partial_t \mathsf{q}[p'] + \partial_x \mathsf{j}[p'] + \mathsf{j}[p', h_0] = 0$ and $\partial_t \mathsf{q}[E'] + \partial_x \mathsf{j}[E'] + \mathsf{j}[E', h_0] = 0$, and using the following identities (see (12) and (28)):

$$\mathsf{j}[E', h_0] = \mathsf{j}[p, h_0] = (p', h_0) = \mathsf{q}_0 \tag{62}$$

and

$$\mathsf{j}[p', h_0] = (p'', h_0) = \begin{cases} 0 & \text{(Galilean)} \\ (E', h_0) = \mathsf{j}_0 & \text{(relativistic)}, \end{cases} \tag{63}$$

we find

$$\partial_t v + v \partial_x v = -\frac{1}{\rho_{\text{fl}}} \partial_x \mathscr{P} - \partial_x V \left( \frac{\mathsf{q}_0}{\rho_{\text{fl}}} - \begin{cases} 0 & \text{(Galilean)} \\ v \mathsf{j}_0 & \text{(relativistic)} \end{cases} \right), \tag{64}$$

where $\mathsf{q}_0$ is the charge density and $\mathsf{j}_0$ is the charge current (the densities and current of the charge $Q_0$ associated to the force term). In the Galilean case with a single-particle spectrum (such as the Lieb-Liniger model), we have $\rho_{\text{fl}} = m \mathsf{q}_0$ and thus we find the usual forced Euler equation,

$$\partial_t v + v \partial_x v = -\frac{1}{\rho_{\text{fl}}} \partial_x \mathscr{P} - \frac{\partial_x V}{m} \quad \text{(Galilean, single-particle spectrum)}. \tag{65}$$

So far we have considered only ideal fluids, that do not account for viscosity. An accurate consideration of viscosity terms corresponding to the underlying many-body model requires an analysis of how the unitary dynamics approaches pure hydrodynamics. However, one may consider a simple, possible correction to (14) that could account for the presence of viscosity effects. Let us exemplify in the Galilean case with a single-particle spectrum. From standard hydrodynamic arguments, the Navier-Stokes equation in one-dimensional non-relativistic systems reads

$$\partial_t v + v \partial_x v = -\frac{1}{\rho_{\text{fl}}} \partial_x \mathscr{P} + \zeta \frac{1}{\rho_{\text{fl}}} \partial_x^2 v, \tag{66}$$

where $\zeta$ is the (mass-normalized) bulk viscosity (note that we do not have the kinematic viscosity as there occurs no shear flow in one dimension). A continuity equation for $\rho_{\text{p}}(\boldsymbol{\theta})$ that gives the above Navier-Stokes equation is

$$\partial_t \rho_{\text{p}}(\boldsymbol{\theta}) + \partial_x (v^{\text{dr}}(\boldsymbol{\theta}) \rho_{\text{p}}(\boldsymbol{\theta})) = \zeta \partial_x^2 \left( \frac{\rho_{\text{p}}(\boldsymbol{\theta})}{\rho_{\text{fl}}} \right). \tag{67}$$

This might or might not correspond to any underlying quantum model, but in any case it could provide a way of regularizing the GHD equations for numerical purposes.[11] It would be interesting to analyze further such viscosity corrections.

## 7  Conclusions

In this letter, we further developed the generalized hydrodynamics (GHD) theory first proposed in [19]. We showed that the GGE equations of state, at the basis of GHD, follow from a principle of hydrodynamic conservation of entropy. We provide in Appendix A arguments for the emergence of GHD in general free-particle relativistic models under smoothness assumptions, which we expect could be extended to interacting models using the form factor program. Then, we generalized to flows generated by arbitrary conserved charges, and employed this in order to establish the conservation equations (46), (47) and continuity equations (48), (49) within an external field, be it a force field, a temperature field or any other field associated to conserved quantities of the model. We expect that these equations should effectively capture the large-scale (long-wavelength) dynamics of a Lieb-Liniger model in an external potential, such as a harmonic potential (see e.g. [43]). This, we believe, is particularly interesting: indeed, despite a lack of full justification, conventional hydrodynamics has been exploited in

---

[11]We remark that this correction does not applied to noninteracting models, such as the free Galilean fermion model where no viscosity term is admitted.

analyzing the quench dynamics of one-dimenional bose gases in a trap potentials [44–46], and we believe our equations might lead to more accurate results. In particular, the consideration of all conservation laws in the force-field GHD might give rise to a more accurate theoretical description of the notable "quantum Newton's cradle" experiment [47]. All equations hitherto derived within GHD are, however, for ideal fluids: no dissipation effect has been taken account of. For a precise treatment one has to add viscosity terms. We proposed one possibility from considering the Navier-Stokes equation, but we expect a more in-depth study will be necessary in order to clarify this aspect.

# 8 Acknowledgement

TY and BD are indebted to B. Bertini, M. Fagotti and especially J. Dubail for a careful reading of the manuscript and for discussions. BD thanks F. Essler, as well as the group of Statistical Physics at the Scuola Internazionale Superiore di Studi Avanzati (SISSA), Trieste, Italy, for discussions. TY and BD are grateful to J.-S. Caux for sharing ideas during the conference "Entanglement and non-equilibrium physics of pure and disordered systems", International Centre for Theoretical Physics (ICTP), Trieste, Italy. BD thanks H. Spohn for discussions during the conference "Non-equilibrium dynamics in classical and quantum systems: from quenches to slow relaxations", Pont-à-Mousson, France. TY acknowledges support from the Takenaka Scholarship Foundation for a scholarship. BD acknowledges support from SISSA, ICTP, the University of Bologna and the University of Lorraine where parts of this work were done.

# A Emergence of GGE equations of state in free-particle models

The problem of showing the emergence of hydrodynamics in many-body systems is notoriously difficult, see [40, 41] for recent progress. This is particularly true because usual hydrodynamics requires, as per its principles, strong interactions, by their nature hard to treat analytically. The interaction should provide the mixing necessary in order for all degrees of freedom that do not follow a conservation law to thermalize; thus minimizing, locally, the free energy under the conditions of all local conservation laws, and rendering applicable, locally, the thermal equations of state. As explained in [19], the sole assumption at the basis of GHD is, likewise, the emergence, in a uniform enough fashion and at large enough times, of the GGE equations of state at every point in space-time. GHD follows from this, simply by combining it with the conservation equations (1) of the model's unitary dynamics. In this respect, GHD offers a unique opportunity in that it accounts for infinitely-many conservation laws: as a consequence much less interaction effects, or mixing, is required for the emergence of the GGE equations of state. This is particularly evident in "quadratic models", or models whose asymptotic particles do not interact. In such models, GGE equations of states should still emerge, although the interaction between fundamental degrees of freedom is quadratic and amenable to exact treatment. Thus, in these models, we may analyze with much more depth these fundamental principles, making use of the large simplification afforded by the triviality of the scattering matrix.

An important question is therefore what basic properties either of the initial state or of the large-time evolution guarantee that the GGE equations of state emerge in free-particle models. Although hydrodynamic ideas have been used successfully in the past in such models [27–32], to our knowledge, no general assessment of such conditions for the emergence of GGE equations of states, or of hydrodynamcis, have been provided.[12] In this section we propose

---

[12]It is also an interesting question to connect the free-particle hydrodynamics developed in past works with the free-particle specialization of the present GHD. However we keep this question for future works.

such conditions. We provide arguments to the effect that, under homogeneous time evolution, if densities and currents become, at long times, smooth enough in space-time, with a variation scale growing unboundedly with time, then the GGE equations of state and GHD emerge. In other words, we show that GGE equations of state hold in homogeneous, stationary states; and if the size of fluid cells, wherein uniform near-homogeneity and near-stationary hold, grow with time, then GGE equations of states are approached and GHD becomes increasingly accurate.

A free-particle model is characterized by the fact that $\varphi(\boldsymbol{\theta}) = 0$. For simplicity and clarity, in the following we specialize to the case of a single relativistic particle, but the derivation below can be generalized straightforwardly (to many particles, and to other dispersion relations). Let us therefore consider some initial state $\langle \cdots \rangle$, and let us evaluate in this state observables evolved in time:

$$\langle \mathcal{O}(x,t) \rangle = \langle e^{iHt} \mathcal{O}(x) e^{-iHt} \rangle. \tag{68}$$

Of course, it cannot be expected in general that GGE equations of state emerge for any initial state, as cases where hydrodynamics fail certainly exist. Hence we need a condition which will guarantee that such "pathological" cases are avoided. A natural condition is the requirement that the long-time limit be smooth enough.

We first assume that everywhere in space-time (at positive times), average densities and currents $\langle \mathcal{O}(x,t) \rangle$ stay uniformly finite. Let us also assume that, as time $t$ becomes large, and uniformly within some region $\mathcal{R}$ of space-time that is unbounded in the positive time direction, average densities and currents display order-1 variations in space-time on lengths scales that diverge as $t$ grows. We express this latter assumption more precisely by considering averages over Gaussian cells centered at $x, t$ of extent $T = T(t)$:

$$\bar{\mathcal{O}}(x,t;\lambda) = \frac{1}{2\pi\lambda^2 T^2} \int d\tau dy \, e^{-\frac{r^2}{\lambda^2 T^2}} \langle \mathcal{O}(y,\tau) \rangle, \tag{69}$$

where $r = \sqrt{(y-x)^2 + (\tau-t)^2}$. Then the assumption is that there is a $T = T(t)$ growing unboundedly with time, such that

$$\lim_{\lambda \to 0} \bar{\mathcal{O}}(x,t;\lambda) = \langle \mathcal{O}(x,t) \rangle \quad \text{uniformly on } (x,t) \in \mathcal{R}. \tag{70}$$

For any finite $(x,t)$, it is clear that the limit is as above; the assumption is that this holds uniformly in $\mathcal{R}$, this being most nontrivial in the long-time subregion of $\mathcal{R}$.

Then, under this assumption, we argue in below that the GGE equations of state emerge uniformly at long times in $\mathcal{R}$.[13] In order to make this conclusion more precise, recall that the averages of conserved densities $q[h]$ and currents $j[h]$ associated to one-particle eigenvalue $h(\theta)$ are linear functionals of $h$ as per (5):

$$\langle q[h](x,t) \rangle = \int d\theta \, \rho_p(\theta;x,t) h(\theta)$$
$$\langle j[h](x,t) \rangle = \int d\theta \, \rho_c(\theta;x,t) h(\theta). \tag{71}$$

For a generic state and generic $x,t$, the densities $\rho_p(\theta;x,t)$ and $\rho_c(\theta;x,t)$ are functionally not related to each other. The emergence of the GGE equations of state is the statement of the emergence of the relation (8), or equivalently (6) with (9) (or (13)). In free relativistic particle models, this is particularly simple as the effective velocity is the group velocity $v^{gr}(\theta) = \tanh\theta$:

---

[13] In the present discussion, we do not discuss conditions of uniformness in $\theta$ or in $h(\theta)$ that might be necessary in order to go between quasi-particle quantities and local observables.

the relation is $\rho_c(\theta; x, t) - v^{gr}(\theta)\rho_p(\theta; x, t) = 0$. We show that this relation emerges uniformly in the region $\mathscr{R}$ as $t \to \infty$:

$$\lim_{\tau \to \infty} \sup\left(\rho_c(\theta; x, t) - v^{gr}(\theta)\rho_p(\theta; x, t) : (x, t) \in \mathscr{R}, t > \tau\right) = 0 \tag{72}$$

A sketch of the proof is as follows. Let $j[h](x, t)$ be the current associated to the GGE determined by quasi-particle density $\rho_p(\theta; x, t)$. Then this implies that the current difference $\langle j[h](x, t)\rangle - j[h](x, t))$ goes to zero uniformly as above. This gives rise to the integral form of conservation equations, with uniform correction terms that are smaller than the total length of the path:

$$\int_{x_1}^{x_2} dx\, (q[h](x, t_2) - q[h](x, t_1)) + \int_{t_1}^{t_2} dt\, (j[h](x_2, t) - j[h](x_1, t)) = o\left(|x_2 - x_1| + |t_2 - t_1|\right) \tag{73}$$

(as $t_1, t_2 \to \infty$ and uniformly for $(x_1, t_1)$ and $(x_2, t_2)$ inside $\mathscr{R}$). We therefore conclude that the integral form of the conservation equations on finite paths, up to $o(1)$ corrections, holds for the *scaled quantities* $\tilde{q}[h](x, t) = q[h](\lambda x, \lambda t)$ and $\tilde{j}[h](x, t) = j[h](\lambda x, \lambda t)$, for any scale $\lambda$ that diverges with time. With $\lambda \propto T$, these scaled quantities have $O(1)$ variations on $O(1)$ lengths, and are the hydrodynamic variables; the scaling with $\lambda$ emulates the taking of large fluid cells (and often one may take $T(t) = t$, so that fluid cells grow linearly with time). We therefore find the emerging hydrodynamic conservation equations, in integral form, for hydrodynamic variables. Assuming differentiability, this implies the differential form (3).

We finally note that we may apply the above result to the case where the state is stationary and homogeneous. In this case, it is clear that the assumption is fulfilled, and we conclude that in such states, be them GGE states or not, averages of local densities and currents *must* be reproducible by a GGE.

In a free particle model, average densities and currents take are bilinears in terms of canonical annihilation and creation operators $A(\theta), A^\dagger(\theta)$. Therefore, they take the following general form

$$\langle q[h](x, t)\rangle = \int d\theta_1 d\theta_2\Big(b[h](\theta_1, \theta_2)\langle A_1^\dagger A_2\rangle e^{i(E_1 - E_2)t - i(p_1 - p_2)x}$$
$$+ c[h](\theta_1, \theta_2)\langle A_1 A_2\rangle e^{-i(E_1 + E_2)t + i(p_1 + p_2)x} + h.c\Big) \tag{74}$$

$$\langle j[h](x, t)\rangle = \int d\theta_1 d\theta_2\Big(\tilde{b}[h](\theta_1, \theta_2)\langle A_1^\dagger A_2\rangle e^{i(E_1 - E_2)t - i(p_1 - p_2)x}$$
$$+ \tilde{c}[h](\theta_1, \theta_2)\langle A_1 A_2\rangle e^{-i(E_1 + E_2)t + i(p_1 + p_2)x} + h.c\Big) \tag{75}$$

where $b[h](\theta_1, \theta_2)$, $\tilde{b}[h](\theta_1, \theta_2)$, $c[h](\theta_1, \theta_2)$ and $\tilde{c}[h](\theta_1, \theta_2)$ are linear functionals of $h$ (here and below indices in $A_j$, $E_j$ and $p_j$ represent the rapidity argument $\theta_j$, and $E_j$ is the energy and $p_j$ the momentum). Recall that $\langle \cdots \rangle$ represents the initial state.

In specific models, it is a simple matter to evaluate the coefficients $b[h](\theta_1, \theta_2)$, $\tilde{b}[h](\theta_1, \theta_2)$, $c[h](\theta_1, \theta_2)$ and $\tilde{c}[h](\theta_1, \theta_2)$ explicitly. In some simple free-fermionic models, these coefficients may be simple enough to guarantee that, with Galilean invariance, the hydrodynamic equations hold exactly independently of the initial state and at all times [48]. However, here we leave these coefficients as general as possible, and impose only conditions that arise from general principles.

We may use the fact that $h(\theta)$ is the one-particle eigenvalue in order to have conditions on $b[h](\theta_1, \theta_2)$. For definiteness, consider the normalization $2\pi[A(\theta_1), A^\dagger(\theta_2)] = E(\theta_1)\delta(\theta_1 - \theta_2)$ (where $[\cdot, \cdot]$ is either the commutator or the anti-commutator) and the one-particle states $|\theta\rangle = (2\pi)^{\frac{1}{2}}E(\theta)^{-\frac{1}{2}}A^\dagger(\theta)|vac\rangle$. These have normalization $\langle \theta_1 | \theta_2 \rangle = \delta(\theta_1 - \theta_2)$. In order to

get a condition on $b[h](\theta_1, \theta_2)$, we use the fact that it is independent of the initial state, and choose it of the form $\langle \cdots \rangle = \int d\theta_1 d\theta_2 \, f(\theta_1, \theta_2)\langle \theta_1 | \cdots | \theta_2 \rangle$ with $f(\theta_1, \theta_2)$ smooth and $f(\theta, \theta)$ decaying fast enough at infinity. On one hand, we have

$$\langle A^\dagger(\theta_1)A(\theta_2)\rangle = (2\pi)^{-1}\sqrt{E(\theta_1)E(\theta_2)}f(\theta_1, \theta_2), \tag{76}$$

and $\langle A(\theta_1)A(\theta_2)\rangle = 0$. Evaluating the integral $\langle Q[h]\rangle = \int dx \, \langle q[h](x, 0)\rangle$ using (75) with $t = 0$, we therefore obtain $Q[h] = \int d\theta \, b[h](\theta, \theta)f(\theta, \theta)$. On the other hand, since we know that $Q[h]|\theta\rangle = h(\theta)|\theta\rangle$, we have $\langle Q[h]\rangle = \int d\theta \, f(\theta, \theta)h(\theta)$. Therefore, we must have

$$b[h](\theta, \theta) = h(\theta), \tag{77}$$

and we further assume that $b[h](\theta_1, \theta_2)$ is Taylor expandable around $\theta_1 = \theta_2$ (which is the case in all free models we know).

Further, by the conservation law, it is immediate that

$$\frac{\tilde{b}[h](\theta_1, \theta_2)}{b[h](\theta_1, \theta_2)} = \frac{E_1 - E_2}{p_1 - p_2}, \qquad \frac{\tilde{c}[h](\theta_1, \theta_2)}{c[h](\theta_1, \theta_2)} = \frac{E_1 + E_2}{p_1 + p_2}. \tag{78}$$

We are looking to show (72). This can be written as the statement that

$$\lim_{t \to \infty} \left( \frac{\delta}{\delta h(\theta)}\langle j[h](\xi t, t)\rangle - \tanh(\theta)\frac{\delta}{\delta h(\theta)}\langle q[h](\xi t, t)\rangle \right) = 0, \tag{79}$$

uniformly on $\xi$.

In order to prove this, let us first analyze what uniform finiteness in space-time means for the initial state itself. Assuming that $\langle A_1^\dagger A_2 \rangle = O\left((\theta_1 - \theta_2)^b\right)$ as $\theta_1 \to \theta_2$, we will conclude that we must have $b \geq -1$; the distribution $\langle A_1^\dagger A_2 \rangle$ may also contain a delta-function term of the type $f(\theta_1)\delta(\theta_1 - \theta_2)$ with $f(\theta)$ decaying fast enough at infinity. We consider Gaussian-cell averages of densities, $\bar{q}[h](x, t; \lambda)$ (see (69)) (the same conclusion is obtained using currents instead of densities). This should stay finite, in particular, with $T = t$, $\lambda = 1$ and $x = 0$, as $t \to \infty$. We use

$$\frac{1}{\sqrt{2\pi T}}\int_{-\infty}^{\infty} d\tau \, e^{-\frac{(\tau - t)^2}{2T^2} + i\tau \mathscr{E}} = e^{it\mathscr{E} - T^2\mathscr{E}^2/2}, \tag{80}$$

and similarly for the $y$ integral in (69), as well as the mode expansion (75). We see, from the fact that $E_1 + E_2$ is always positive, that all terms in (75) involving $\langle A_1 A_2 \rangle$ and its hermitian conjugate will have exponentially decaying contributions as $t \to \infty$. The remaining terms are

$$\int d\theta_1 d\theta_2 \, b[h](\theta_1, \theta_2)e^{iE_{12}t - t^2(E_{12}^2 + p_{12}^2)/2}\langle A_1^\dagger A_2 \rangle, \tag{81}$$

where $p_{12} := p_1 - p_2$ and $E_{12} := E_1 - E_2$. We recall that $b[h](\theta_1, \theta_2)$ is regular at $\theta_1 = \theta_2$. We further assume that it behaves well enough at large rapidities, so that we do not worry about the large-rapidity region of the integrals.

The eventual delta-function term in $\langle A_1^\dagger A_2 \rangle$ leads to a finite contribution to (81) by our assumptions concerning behaviors at infinite rapidities. On the other hand, at large $t$, the algebraic contribution of $\langle A_1^\dagger A_2 \rangle$ to (81) can be analyzed by a stationary phase argument. Setting $u := p_2$ and $w := p_2^2 + E_2^2$, the stationary phase occurs at

$$\theta_1 - \theta_2 =: \theta_{12} = \theta^\star := iu/(wt) + O(1/t^2). \tag{82}$$

Keeping only the terms up to quadratic order in $\theta_{12} - \theta^\star$ in the exponential and using that $\langle A_1^\dagger A_2 \rangle = O\left((\theta_{12})^b\right)$ we are left with

$$\sim \int d\theta_1 d\theta_2 \, O(\theta_{12}^b)e^{-\frac{w(t^2 + O(t))}{2}\left(\theta_{12} - \frac{iu}{wt} + O\left(\frac{1}{t^2}\right)\right)^2 - \frac{u^2}{2w} + O\left(\frac{1}{t}\right)} = \int d\theta_2 \, O\left(\frac{1}{t^{1+b}}\right). \tag{83}$$

Finiteness thus requires $b \geq -1$.

Now consider

$$\frac{\delta}{\delta h(\theta)} \bar{j}[h](x,t;\lambda) - \tanh(\theta) \frac{\delta}{\delta h(\theta)} \bar{q}[h](x,t;\lambda) \tag{84}$$

for some $T = T(t)$ in (69) that grows unboundedly with $t$. Again, we see that all terms in (75) involving $\langle A_1 A_2 \rangle$ and its hermitian conjugate will have exponentially decaying contributions in (84) as $t \to \infty$. Terms involving $\langle A_1^\dagger A_2 \rangle$, on the other hand, are of the form

$$\int d\theta_1 d\theta_2 \left( \frac{E_1 - E_2}{p_1 - p_2} - v^{gr}(\theta) \right) \frac{\delta}{\delta h(\theta)} b[h](\theta_1, \theta_2) \times e^{iE_{12}t - ip_{12}x - T^2(E_{12}^2 + p_{12}^2)/2} \langle A_1^\dagger A_2 \rangle. \tag{85}$$

We may bound this integral by replacing the oscillatory factor $e^{iE_{12} - ip_{12}x}$ by 1. At large $T$ this can then be analyzed by a stationary phase argument. The position of the stationary phase is exactly $\theta_1 = \theta_2$, hence the main contribution occurs around $\theta_1 \approx \theta_2$. Thanks to (77), we find

$$\begin{aligned}
\left( \frac{E_1 - E_2}{p_1 - p_2} - v^{gr}(\theta) \right) \frac{\delta}{\delta h(\theta)} b[h](\theta_1, \theta_2) &= \left( \frac{E_1 - E_2}{p_1 - p_2} - v^{gr}(\theta) \right) \left( \delta(\theta - \theta_2) + O(\theta_{12}) \right) \\
&= \left( v^{gr}(\theta_2) - v^{gr}(\theta) + O(\theta_{12}) \right) \left( \delta(\theta - \theta_2) + O(\theta_{12}) \right) \\
&= O(\theta_{12}).
\end{aligned} \tag{86}$$

Therefore, the delta-function part of $\langle A_1^\dagger A_2 \rangle$ does not contribute to the integral (85), and the algebraic contribution becomes, as $t \to \infty$,

$$\leq \int d\theta_1 d\theta_2 \, O(\theta_{12}^{b+1}) e^{-\frac{wT^2}{2}\theta_{12}^2} = \int d\theta_2 \, O\left( \frac{1}{T^{b+2}} \right). \tag{87}$$

This is clearly uniform on $(x,t) \in \mathscr{R}$. Since $b \geq -1$, as a consequence, we have found that

$$\lim_{t \to \infty} \left( \frac{\delta}{\delta h(\theta)} \bar{j}[h](x,t;\lambda) - \tanh(\theta) \frac{\delta}{\delta h(\theta)} \bar{q}[h](x,t;\lambda) \right) = 0 \tag{88}$$

uniformly on $\mathscr{R}$. By the assumption (70), this is sufficient to show (79).

This is of course far from being a complete or rigorous proof. For instance, we have omitted the discussion of how the assumption (70) is uniform with respect to the observables $\mathcal{O}$ themselves (allowing us to take $h(\theta)$-derivatives). We have also omitted the detailed dependencies on $\theta_1, \theta_2$ in expressions of the form $O(\theta_{12}^c)$, while these are important to make sure that the rapidity integrals are finite. In addition, of course, the stationary phase arguments, while treated with some care, would need to be developed in order to become rigorous. Nevertheless, we believe this provides the main arguments, and shows how GGE equations of state may indeed emerge.

# B  Derivation of hydrodynamic equations within inhomogeneous fields

In order to describe the first part of the result, equation (42), consider the conservation law of the conserved density $q_i$ with respect to the time evolution generated by a conserved quantity $Q_k$,

$$i[Q_k, q_i] + \partial_x j_{k,i} = 0. \tag{89}$$

GGE averages of the associated currents can be evaluated using (28) as $j_{k,i} = j[h_k, h_i]$, which, thanks to (28), takes the explicit form

$$j_{k,i} = \int \frac{d\boldsymbol{\theta}}{2\pi} h'_k(\boldsymbol{\theta}) n(\boldsymbol{\theta}) h_i^{dr}(\boldsymbol{\theta}). \tag{90}$$

Equation (47) (which implies (42)) is shown as follows. Locality of densities imply that there exists a field $\mathcal{O}_{j,i}(x, y)$ supported at $x = y$ (i.e. local at this position) such that

$$i[q_k(y), q_i(x)] = \mathcal{O}_{k,i}(y, x). \tag{91}$$

Since $q_j(x)$ and $q_i(x)$ are local conserved densities, they are not affected by any nontrivial renormalization, and therefore $\mathcal{O}_i(x, y)$ can be written as a finite sum of terms with increasing derivatives of the delta function,

$$\mathcal{O}_{k,i}(y, x) = \sum_{\ell=0}^{L} \mathcal{O}_{k,i;\ell}(x) \delta^{(\ell)}(y - x) \tag{92}$$

where $\mathcal{O}_{k,i;\ell}(x)$ are local fields. Integrating over $y$, by (89) we find that

$$\mathcal{O}_{k,i;0}(x) = -\partial_x j_{k,i}(x). \tag{93}$$

On the other hand, integrating over $x$, we obtain

$$-i[Q_i, q_k(y)] = \sum_{\ell=0}^{L} \partial_y^\ell \mathcal{O}_{k,i;\ell}(y) \tag{94}$$

and therefore comparing with (89) we can make the following identification, using the fact that the only local fields whose derivative is zero are those proportional to the identity:

$$\mathcal{O}_{k,i;1}(y) = j_{i,k}(y) + j_{k,i}(y) - \partial_y \mathcal{Q}_{k,i}(y) - A_{k,i} \mathbf{1} \tag{95}$$

where $\mathcal{Q}_{k,i}(y) := \sum_{\ell=2}^{L} \partial_x^{\ell-2} \mathcal{O}_{k,i;\ell}(y)$.

Here $A_{k,i} = A_{i,k}$ is a constant. It can be seen to vanish as follows. We write it as the following quantity, involving an averages $\langle \cdots \rangle$ in any GGE:

$$A_{i,k} = \int dx \left( ix \langle [q_k(x), q_i(0)] \rangle + j_{i,k} + j_{k,i} \right). \tag{96}$$

By symmetry, this constant is zero whenever $q_k$ and $q_i$ are both parity symmetric or parity anti-symmetric, or whenever their combined transformation under some internal symmetry is nontrivial. One can argue this constant should in fact be identically zero as follows. Note that $A_{i,k}$ is a bilinear functional of $h_i$ and $h_k$, that is $A_{i,j} = A[h_i, h_k]$. Let us consider

$$A[h, g] = \int dx \left( ix \langle [q[g](x), q[h](0)] \rangle + j[h, g] + j[g, h] \right), \tag{97}$$

for functions $h(\boldsymbol{\theta})$ and $g(\boldsymbol{\theta})$ that decay fast enough at infinite rapidities. Let us also consider the GGE $\langle \cdots \rangle$ to be a thermal state in the limit of large temperatures. In this limit [36], the occupation number $n(\boldsymbol{\theta})$ has a large flat plateau, and decays to zero beyond this plateau. The regions where it starts decaying to zero are further and further away from $\theta = 0$ as the large temperature limit is taken. Therefore, in (11) and (10), for $h, g$ as above, we may consider

$n(\boldsymbol{\theta})$ to be a constant, independent of the rapidity. Hence by integration by parts, we have $(h')^{\mathrm{dr}} = (h^{\mathrm{dr}})'$. Thus, using (28) and (10) (and its symmetry property), we have

$$
\begin{aligned}
\mathsf{j}[h,g] = (h',g) &= \int \frac{\mathrm{d}\boldsymbol{\theta}}{2\pi} h'(\boldsymbol{\theta}) g^{\mathrm{dr}}(\boldsymbol{\theta}) \\
&= -\int \frac{\mathrm{d}\boldsymbol{\theta}}{2\pi} h(\boldsymbol{\theta}) (g^{\mathrm{dr}}(\boldsymbol{\theta}))' \\
&= -\int \frac{\mathrm{d}\boldsymbol{\theta}}{2\pi} h(\boldsymbol{\theta}) (g')^{\mathrm{dr}}(\boldsymbol{\theta}) \\
&= -(h,g') = -(g',h) = -\mathsf{j}[g,h].
\end{aligned}
$$

That is, in this limit $\mathsf{j}[h,g] + \mathsf{j}[g,h] = 0$. Further, in the infinite temperature limit the state is the trace state, which has the cyclic property $\langle AB \rangle = \langle BA \rangle$. As a consequence[14], in this limit $\langle [q[g](x), q[h](0)] \rangle = 0$. Therefore, since $A[h,g]$ is independent of the state, we must have $A[h,g] = 0$. We thus conclude that this is the zero bilinear functional, and thus $A_{i,k} = 0$ for all $i$ and $k$.

Note that one can further check that the result (95) with $A_{k,i} = 0$ agrees, in the case where $q_i$ and $q_k$ are either energy or momentum densities, with the first-derivative terms of the commutators of the stress-energy tensor calculated in [49].

We can then compute the time evolution within the inhomogeneous field as follows:

$$
\begin{aligned}
\mathrm{i}[H_{\mathrm{field}}, q_i(x)] = \mathrm{i}[H, q_i(x)] &+ \mathrm{i}\sum_k \int \mathrm{d}y\, V_k(y)[q_k(y), q_i(x)] \\
&= -\partial_x j_i(x) + \sum_k \sum_{\ell=0}^{L} (-\partial_x)^\ell V_k(x) \mathscr{O}_{k,i;\ell}(x) \\
&= -\partial_x j_i(x) - \sum_k \big( \partial_x (V_k(x) j_{k,i}(x)) + j_{i,k}(x) \partial_x V_k(x) \big) + \dots \\
&= -\partial_x j_i(x) - \big( \partial_x (\mathsf{j}[W(x), h_i](x)) + \mathsf{j}[h_i, \partial_x W(x)](x) \big) + \dots
\end{aligned}
\tag{98}
$$

where $W(x) = \sum_k V_k(x) h_k$ is the one-particle external-field function (for every $x$, it is a function of $\boldsymbol{\theta}$). We have used integration by parts, assuming that boundary terms at infinity do not contribute. The terms omitted are "higher-derivative terms": they are composed of products of the first or higher derivative of the potentials $V_k(x)$ times local fields and their derivatives, with, in total, two or more space derivatives.

Integrating over a large space-time cell $\Omega$, we obtain the integral form of a conservation equation, $\int_{\partial\Omega} \mathrm{d}\vec{x} \wedge \vec{q}(\vec{x}) = S_\Omega$, $\vec{x} = (x,t)$, $\vec{q} = (q,j)$, for the density $q = q_i$ and the modified current $j = j_i + \mathsf{j}[W, h_i]$, with sources within the cell, $S_\Omega = \int_\Omega \mathrm{d}x \mathrm{d}t\, \mathsf{j}([h_i, \partial_x W](x,t)$. We may now make the hydrodynamic assumption that averages of local observables are evaluated in local GGEs, and reverting to the differential form of this conservation equation, this shows (47).

In a pure force field, i.e. with $W(x)' = 0$, the equation simplifies, since in this case $\mathsf{j}[W(x), h_i](x) = 0$. For evolution within a pure force field, we are therefore left with

$$
\mathrm{i}[H_{\mathrm{force}}, q_i(x)] = -\partial_x j_i(x) - \mathsf{j}[h_i, \partial_x W(x)](x) + \dots
\tag{99}
$$

---

[14]Taking the infinite-temperature limit in QFT is delicate, as large temperatures bring the system much beyond the quantum critical point. However, choosing $h$ and $g$ to decay at large rapidities amounts to a UV regularization of the fields $q[h](x)$ and $q[g](x)$ (which are therefore not local anymore). This UV regularization guarantees that the energy scale of the temperature, in the large-temperature limit, is beyond the UV scale of the observables, and thus the limit is indeed described by the microscopic formula, which is a trace state.

which implies (42).

In order to show (48) and (49), Equation (98) is written, using TBA and in particular using (90) and the symmetry of the bilinear form (10), as

$$0 = \int \frac{\mathrm{d}\boldsymbol{\theta}}{2\pi} \Big[ 2\pi h_i \big( \partial_t \rho_\mathrm{p} + \partial_x \big( v^\mathrm{eff} \rho_\mathrm{p} \big) \big) + \sum_k \big( h_i \partial_x (V_k \, n (h'_k)^\mathrm{dr}) + (\partial_x V_k) \, n h_k^\mathrm{dr} h'_i \big) \Big], \qquad (100)$$

(here for lightness of notation, we omit the explicit $\boldsymbol{\theta}$ and $x$ dependences, and recall that primes (') indicate $\theta$-derivatives). Using integration by parts for the last term in the square brackets, and using the fact that this holds for every function $h_i$ (assuming completeness of this space of functions), we obtain

$$2\pi \big( \partial_t \rho_\mathrm{p} + \partial_x \big( v^\mathrm{eff} \rho_\mathrm{p} \big) \big) + \sum_k \big( \partial_x (V_k \, n (h'_k)^\mathrm{dr}) - \partial_x V_k \, (n h_k^\mathrm{dr})' \big) = 0. \qquad (101)$$

Let us use integral-operator notations, with measure $\int \mathrm{d}\boldsymbol{\theta}/(2\pi)$. Consider the diagonal operator $\mathcal{N}$ with kernel $\mathcal{N}(\boldsymbol{\theta}, \boldsymbol{\alpha}) = 2\pi \, n(\boldsymbol{\theta}) \delta(\theta - \alpha) \delta_{a,b}$, the vectors $p'$, $E'$ and $h_0$ with elements $p'(\boldsymbol{\theta})$, $E'(\boldsymbol{\theta})$ and $h_0(\boldsymbol{\theta})$ respectively, and the operator $\varphi$ with kernel $\varphi(\boldsymbol{\theta}, \boldsymbol{\alpha})$. Then

$$2\pi \rho_\mathrm{p} = \mathcal{N}(1 - \varphi \mathcal{N})^{-1} p'$$
$$2\pi v^\mathrm{eff} \rho_\mathrm{p} = \mathcal{N}(1 - \varphi \mathcal{N})^{-1} E'$$
$$n h_k^\mathrm{dr} = \mathcal{N}(1 - \varphi \mathcal{N})^{-1} h_k$$
$$n(h'_k)^\mathrm{dr} = \mathcal{N}(1 - \varphi \mathcal{N})^{-1} h'_k. \qquad (102)$$

Using the first, second and last of these relations, as well as (30), we see that we can combine the term $\partial_x (v^\mathrm{eff} \rho_\mathrm{p})$ with $\partial_x (V_k \, n (h'_k)^\mathrm{dr})$ into $\partial_x (v^\mathrm{eff} [E + W] \rho_\mathrm{p})$. On the other hand, using the first and the third, as well as (50), we see that $\sum_k \partial_x V_k (n h_k^\mathrm{dr})' = 2\pi \partial_\theta (a^\mathrm{eff} \rho_\mathrm{p})$. Therefore, this indeed reproduces (49).

Next we derive (48). Note that $\mathcal{N}(1 - \varphi \mathcal{N})^{-1} = \mathcal{N} + \mathcal{N} \varphi \mathcal{N} + \mathcal{N} \varphi \mathcal{N} \varphi \mathcal{N} + \dots$. Differentiating with respect to any internal parameter (say $u = x$ or $u = t$) that $\varphi$ does not depend on, we have

$$\partial_u \big( \mathcal{N} \varphi \mathcal{N} \varphi \cdots \big) = \big( \partial_u \mathcal{N} \big) \varphi \mathcal{N} \varphi \cdots + \mathcal{N} \varphi \big( \partial_u \mathcal{N} \big) \varphi \cdots + \dots \qquad (103)$$

Therefore, it is seen that

$$\partial_u \big( \mathcal{N}(1 - \varphi \mathcal{N})^{-1} \big) = (1 - \mathcal{N} \varphi)^{-1} \big( \partial_u \mathcal{N} \big) (1 - \varphi \mathcal{N})^{-1}. \qquad (104)$$

Similarly, in order to differentiate with respect to $\theta$ we may use integration by parts, along with the fact that $\varphi$ depends on the difference of rapidities. Explicitly, we have for instance

$$\partial_\theta \left( \int \mathrm{d}\boldsymbol{\theta}' \, n(\boldsymbol{\theta}) \varphi(\boldsymbol{\theta}, \boldsymbol{\theta}') n(\boldsymbol{\theta}') h_k(\boldsymbol{\theta}') \right) = \int \mathrm{d}\boldsymbol{\theta}' \, \partial_\theta n(\boldsymbol{\theta}) \varphi(\boldsymbol{\theta}, \boldsymbol{\theta}') n(\boldsymbol{\theta}') h_k(\boldsymbol{\theta}')$$
$$+ \int \mathrm{d}\boldsymbol{\theta}' \, n(\boldsymbol{\theta}) \partial_\theta \varphi(\boldsymbol{\theta}, \boldsymbol{\theta}') n(\boldsymbol{\theta}') h_k(\boldsymbol{\theta}'), \quad (105)$$

and the last term can be written as

$$\int \mathrm{d}\boldsymbol{\theta}' \, n(\boldsymbol{\theta}) \varphi(\boldsymbol{\theta}, \boldsymbol{\theta}') \partial_{\theta'} \big( n(\boldsymbol{\theta}') h_k(\boldsymbol{\theta}') \big).$$

Hence,

$$\big( \mathcal{N} \varphi \mathcal{N} h_k \big)' = \mathcal{N}' \varphi \mathcal{N} h_k + \mathcal{N} \varphi \mathcal{N}' h_k + \mathcal{N} \varphi \mathcal{N} h'_k. \qquad (106)$$

Generalizing to all orders, this gives

$$\left(\mathcal{N}(1-\varphi\mathcal{N})^{-1}h_k\right)' = (1-\mathcal{N}\varphi)^{-1}\left(\mathcal{N}'\right)(1-\varphi\mathcal{N})^{-1}h_k + n(h_k')^{\mathrm{dr}}. \tag{107}$$

Writing $\partial_x(V_k\, n(h_k')^{\mathrm{dr}}) = \partial_x V_k\, n(h_k')^{\mathrm{dr}} + V_k\, \partial_x(n(h_k')^{\mathrm{dr}})$, the last term in the equation above cancels one of the terms in the summand in (101). The summand in (101) therefore simplifies to

$$\partial_x V_k\,(1-\mathcal{N}\varphi)^{-1}\left(\mathcal{N}'\right)(1-\varphi\mathcal{N})^{-1}h_k + V_k\partial_x(n(h_k')^{\mathrm{dr}}).$$

We may evaluate the last term in this expression, as well as the derivatives of $\rho_{\mathrm{p}}$ and $v^{\mathrm{eff}}\rho_{\mathrm{p}}$ in (101), using (104). Premultiplying by $(1-\mathcal{N}\varphi)$ in order to cancel the common operatorial factor, and then multiplying by $2\pi$ and dividing by $(p')^{\mathrm{dr}}$, we obtain the following:

$$\partial_t n(\boldsymbol{\theta}) + v^{\mathrm{eff}}[E+W](\boldsymbol{\theta})\,\partial_x n(\boldsymbol{\theta}) + a^{\mathrm{eff}}(\boldsymbol{\theta})\,\partial_\theta n(\boldsymbol{\theta}) = 0. \tag{108}$$

This indeed reproduces (48).

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
