# Peer review of "A note on generalized hydrodynamics: inhomogeneous fields and other concepts"

_SciPost Physics, doi:SciPost Phys. 2, 014 (2017)_

## Round 1 · Referee Report · Anonymous · 2017-3-3

Strengths
1- interesting problem
2- timely subject
3- self-contained paper
Weaknesses
1- some parts are difficult to read
2- some underlying assumptions are not stated clearly
3- there are no explicit examples confirming the validities of the equations
Report
The authors study the interesting problem of the emergence of a generalised hydrodynamic description in “quasi-integrable” models. Specifically, they propose a generalisation of the results obtained in Refs [19,20] which takes care of weak Hamiltonian inhomogeneities. In addition, they review and elaborate on the previous results, including some interesting insights. For example, they show that it is possible of replace the TBA equation satisfied by the velocity (entering the continuity equation) by postulating the local conservation of entropy.
I think that the paper is interesting, however, there are some parts that I don’t fully understand.
Section IV and Appendix A are about the emergence of generalised hydrodynamics in free-particle models. The authors aim at proving some very general statements but, in the end, their analysis seems to rely on some strong assumptions. In particular, I don’t think that the authors adequately cover the types of initial states usually considered in these problems. For example, are the (smoothness) assumptions of Appendix A satisfied in domain-wall problems (considered by the authors in their prequel)?
Section V aims at describing “how to represent the dynamics associated to all conservation laws”. As far as I can see, there is an implicit assumption that the authors overlooked, i.e., the set of the operators commuting with a given conservation law is supposed not to depend on the conservation law itself. In fact, sufficiently simple conserved quantities can have larger sets of conservation laws. $Q_0$ (particle number), introduced in Section VI, constitutes an example. In such cases, a description in terms of particle densities could be insufficient.
In section VI the authors derive the generalised hydrodynamic equations in the presence of external inhomogeneous fields. I like the generality of the presentation but, again, it makes it very difficult to see whether there are implicit assumptions. One possible problem that I see is that the authors are implicitly assuming that the effects of integrability breaking are subleading with respect to the corrective terms they identify. The paper does not propose either a mathematical or a physical justification, so the reader is not put in the condition of perceiving the accuracy of the proposed description. This is a serious problem, since there are no examples that confirm the validity of the equations.
In conclusion, I think that this work deserves publication, but the authors should state the underlying assumptions more clearly and, possibly, include illustrative examples.
Requested changes
1- The sentence before eq. (5) begins with “In any state, GGE or other”; probably the authors mean “In any stationary state, …”.
2- In general, I suggest either to add a physical justification of every assumption, or to provide explicit examples that satisfy the hypotheses.
3- I don't fully understand the central paragraph above eq. (57). Is there any evidence that the late-time stationary state should be thermal? While this is usually expected in homogeneous systems, in the presence of inhomogeneities such statement is highly nontrivial.
4- I don't understand the discussion below eq. (B10). Is the assumption of local GGEs used to remove the higher-derivative terms?
5- The reading of the Appendices is made more difficult by the presence of many references to equations of the main text. The authors should find a solution.
6- Some typos: eq. (11), below eq. (B7), between eq. (B14) and eq. (B15).

---

## Round 1 · Referee Report · Anonymous · 2017-3-3

Strengths
1- The problem studied is currently attracting large interest;
2- The paper gives many non-trivial extensions to the recently developed "generalized hydrodynamics" theory;
Weaknesses
1- Some of the assumptions and the ranges of validity of the results should be clarified;
Report
The goal of the paper is to extend the "generalized hydrodynamics" theory [19], which describes the large time dynamics of integrable models evolving from inhomogeneous initial states. The authors extend the theory in several directions.
i) They show that the continuity equation for the density of particles proposed in Refs. [19] and [20] can be obtained by requiring some entropy conservation principle. More precisely, assuming that the density of entropy (cf. Equation 16) and the density of quasi-particles (cf. Equation 5) satisfy the same continuity equation with an unknown velocity $v^{\rm eff}(\boldsymbol \theta,x,t)$ (fulfilling some conditions), they show that $v^{\rm eff}(\boldsymbol \theta,x,t)$ is the effective velocity of quasi particles.
ii) They (non rigorously) prove, under some assumptions, that the continuity equation(s) emerges from the unitary dynamics of generic free models in the limit $x,t\rightarrow \infty$, with fixed $x/t$.
iii) They consider the evolution generated by a generic conserved quantity and write the corresponding set of continuity equations.
iv) They generalize the continuity equations to describe the time evolution in the presence of inhomogeneous external fields.
v) They show that the continuity equations for some of the conserved quantities can combined to recover Euler-like equations and discuss the possible form of viscous terms.
I think that the paper is interesting. The different extensions to the generalized hydrodynamics considered are relevant and non trivial. The discussion is kept at a very general level and the results can be applied for a wide range of integrable quantum field theories (those with diagonal scattering), such as the Lieb Liniger model or the sinh-Gordon field theory.
In my view, the point of greatest interest among those discussed in the paper is the treatment of inhomogeneous external fields within the hydrodynamic framework. This point represents an important advance because it allows them to quantitatively treat (in the large time limit) some of the systems actually realized in cold atomic experiments. This point of the paper is, however, the one I found more obscure. I think that some of the assumptions and the ranges of validity of the results should be clarified. In particular, if I understand correctly, the derivation of the final equation (Equation (53) or the analogous Equation (54)) goes through two main steps. First the authors derive Equation (51) (of which Equation (47) is a sub-case) using a derivative expansion; to make this step the integrability of the theory with external potentials is not required. Then, they pass from Equation (51) to Equation (53) (or the analogous Equation (54)) where they use a TBA formalism so they approximate the non-integrable theory (an interacting integrable theory becomes generically non-integrable when adding external potentials) with an integrable one. I have two main questions on these steps
1) In which limit does the set of Equations (51) hold? Clearly I expect the equation to be accurate only for large times (as this is true also for the "simple" continuity equation of Refs. [19] and [20]), however, in the "hydrodynamic regime" $x,t\rightarrow\infty$ with fixed $x/t$ the equation does not have a well defined scaling for generic $W(x)$ (the same is true for Equation (47) for generic $V_0(x)$). In the discussion after Equation (56) the authors claim that they expect Equation (53) to hold "... for a finite period of time, whose extent depends on the size of spacial variations of the potentials. Beyond this time, one might expect the integrability-breaking effects of the presence of space varying potentials to become important". Do they expect also Equation (51) to hold in a large time window? If this is the case the effects breaking the validity of (51) can not be integrability-breaking effects since integrability has not been used to derive (51). I think that the authors should discuss this point right after Equation (51).
2) Why can the authors pass from the set of Equations (51) to Equation (53) (or the analogous Equation (54))? Equation (53) does not simply follow from (51), but requires a new level of approximation. For example, taking $V_k(x)=\delta_{k,0}|x|$ I expect Equation (51) to be exact, while Equation (53) can not be exact as the theory will generically be non-integrable. I think that they make some sort of local density approximation, but I could not find any mention to this in the manuscript. The authors should separately discuss the validity of (51) and of (53). Maybe, they could show that Equation (53) works (in some regime) in the simple case of a free theory in an external potential.
I think that these two questions should be addressed and the relevant parts of the paper should be modified before considering it for publication.
In what follows I add a list of minor points/suggestions for the authors.
Requested changes
1- In the discussion after Equation (1), when talking about relevant conserved charges in integrable models, the authors should either cite all the recent fundamental works on that subject or a review;
2- When claiming that Equations (5) describe averages of density and currents "in any state", the authors should specify that the state should be stationary;
3- $E(\boldsymbol \theta)$ and $p(\boldsymbol \theta)$ appearing in Equation (7) are not defined;
4- The assumptions in the proof of Section III should be stated more clearly. I suggest to split the point (i) in more points, as written is difficult to read;
5- In Section VI, the authors in two occasions use $V(x)$ instead of $V_0(x)$ for the "electric potential";
6- Is Equation (44) exact or it is true only at the first order in a derivative expansion? This point is not clear in the current version;
7- Appendix A is very schematic and not easy to read. Moreover, as the authors note at the end, the proof given is not a rigorous one. I suggest changing the title of the appendix in "Sketch of the proof ..." or something on these lines;
8- In the discussion above Equation (A2) the authors write: "Assume that the initial state is of the form $\langle{\ldots}\rangle=\int{\rm d}\theta_1{\rm d}\theta_2 f(\theta_1,\theta_2)\langle{\theta_1|\ldots|\theta_2}\rangle$". I think that they make this assumption only for expectation values of bilinears in $A(\theta)$ and $A^\dagger(\theta)$ (in general it would unreasonably be too stringent). If this is the case they should write it;
9- I find the beginning of Appendix B very confusing. What do the authors mean by "In order to describe the first part of the result, equation (47), ...."? In my understanding Equation (47) is proven as a special case of Equation (52) (as they indeed claim after Equation (B2)). I do not understand why in the first sentence they mention only Equation (47);
10- The labels B9 and B10 refer to the same equation;
11- In Appendix B there are two occurrences of the typo "integration by part" instead of "integration by parts", one after Equation B10 and the other after Equation B12;
Author: Benjamin Doyon on 2017-04-05 [id 112]
(in reply to Report 1 on 2017-03-03)
We are very grateful to both referees for their careful reading of the manuscript and for their suggestions; this helped us a lot to improve the paper.
Main changes:
1) We have reported all of what was previously section IV to appendix A. That is, all discussion of emergence of GGE in free models is in the appendix. We believe this section was not as clear, and that it disrupted a little bit the flow of ideas in the paper. Since it is also not entirely rigorous, we thought it is better to put it in appendix.
2) We have added explanations concerning our assumptions in section V (force fields).
For both referees: Concerning force field (current version Sect V).
The only approximation is the hydrodynamic approximation. This is, in presence of force field, (1) the approximation that local densities and currents are in entropy-maximized states wrt local evolution, and (2) the ``smoothness" approximation which take all $x,t$-dependent quantities, including hydrodynamic fields and force fields, at first derivative order only. It is usually expected, in hydro, that correction terms to the former approximation only lead to second and higher derivatives, so the two approximations are expected to be consistent. That is, it is expected that if things are smooth enough, then locally the state is very near to a homogeneous stationary state wrt the local evolution. The first approximation involves integrability, simply because homogeneous stationary states wrt evolutions by $H+W$ are GGEs. The rest is then purely a derivation, there is no further approximation.
We understand that the referees worry about integrability breaking - clearly $H+\int dx W(x)$ is not integrable. However this does not affect the hydrodynamic description, as this description only relies on the local form of time evolution, which is integrable. An example in ordinary hydro: we know well that in ordinary hydro, a fluid cell is a galilean boost of a thermal state (e.g. it has a local density and a local momentum). This is, for the local state, a homogeneous density matrix $e^{-\beta(H-\nu P)}$ where $P$ is momentum. Now this is true also within force fields such as water in gravitational field. But if the gravitational field is space dependent, it clearly breaks translation invariance. Thus globally only the inhomogeneous hamiltonian is conserved, no momentum. Nevertheless, the fluid cell is still described in the same way, involving $P$. This is because locally, it looks homogeneous, so we recover this conserved quantity. The momentum-breaking nature of the force field appears in the fluid equations themselves, where the force term modifies the macroscopic momentum-conservation equation. It is exactly the same phenomenon here, only extended to all conserved charges. So we do not assume that integrability breaking effects are subleading wrt the terms we identify. All integrability breaking effects, at first-derivative order, are included.
We note additionally: Integrability breaking will be important in time evolution because the higher-derivative terms from force field always play a role - eventually they will make the solution go away from GHD, produce entropy, etc. Thus we would expect, at large times, the system to thermalize according to the (non-integrable) inhomogeneous hamiltonian. But we do not know the time scale for this, and there may be pre-thermalization effects.
We have added two paragraphs on p5, right, which hopefully clarifies our approximations and some of the questions the referees had.
Finally for both reports, concerning examples:
This is of course a very good comment, but unfortunately, in this area of research, nontrivial examples are rather hard to work out fully. This is because hydrodynamics describes very complicated systems in a simple way (this is its strength), so verifying it requires complicated quantum calculations. We have work in progress in this direction, but we believe it is beyond the scope of this paper. As for providing examples with a free theory, again this would be interesting, but we think it's better to study all such examples in a future work dedicated to this.
The point of this paper was to start with the hydrodynamic assumptions only, and get all logical conclusions from this. The only approximations used are those of hydrodynamics, and they have all been stated, we hope clearly in the latest version. The rest is pure derivation from first principles. Since there is no other assumption being made in deriving the main formulae, we believe this is sufficient. Of course, the hydro assumptions, and their consequences, must be checked (and have been checked in the past in ordinary hydro!). But we believe this is for future works.
Report 92:
Concerning the proof in the free-particle case (current version App A).
Hydrodynamics is only supposed to be valid, in general, in smooth enough regions. This is a strong assumption, but in general, Euler-scale hydrodynamics is not expected if this assumption is not satisfied, and so we believe our assumptions are fine in order to study emergence of hydrodynamics.
The initial domain-wall condition is of course not smooth, and hydrodynamics is not expected to correctly describes time evolution from this initial condition. This is known, and in fact has been observed numerically in [20] by comparing with DMRG. However, in [19,20], hydrodynamic is not used from initial time, it is only required at large (in fact infinite) times. Then, the profiles are very smooth, and hydrodynamic holds. In [19,20], the exact solution only requires asymptotics in space, so the precise initial condition is not necessary. In the present manuscript, we show that, if profiles are smooth enough, indeed one can use hydrodynamics, and thus this provides a missing step in [19,20] in the simple case of free models. Of course, free models can be solved by other means, the result here is of conceptual rather than practical interest.
Concerning the conservation laws (Sect V).
It is true that a particular conservation law can have many more commuting charges. However, the main precept of hydrodynamics is that, locally, the system maximize entropy with respect to all commuting charges conserved by the local dynamics. Here, the local dynamics is written in the form $H+W$, where $W$ involves higher conserved charges. So, we must look for the commuting charges with respect to $H+W$. If we take $W = Q_0-H$, then we agree that locally there would be many more conservation laws (the local dynamics, wrt $Q_0$, being trivial). But with $W$ a generic linear combination of higher-spin conserved charges, this is not the case: they have the same set of commuting operators as $H$ itself. This is the basic structure of integrability, any higher-spin conserved quantity can be considered as a hamiltonian, and it has the same set of commuting conserved quantities (same integrability class). This is what we assume: genericity of $W$. In this case, the quasi-particle description is expected to be sufficient (cf [19,20]). Indeed this was not explicitly mentioned, we have added two sentences, page 4, right.
Specific points:
1- No the state is completely general, it does not need to be stationary (or homogeneous). This is because the operator $q$ itself is a linear functional of $h$, hence every matrix element is. Indeed, let $h_i$ be a basis. Then $Q[\sum_i a_i h_i] = \sum_i a_i Q[h_i]$ by linearity. Since $Qh_i = \int dx qh_i$, by locality we must also have $q[\sum_i a_i h_i] = \sum_i a_i q[h_i]$ (this is up to a total derivative of a local field, which can be set to zero by our choice of the density $qh_i$). Thus linearity holds at the operator level, and thus $q(x) = \int d\theta h(\theta) \hat q(x,\theta)$. For the current, this then follows from the general relation between matrix elements of currents and densities (see e.g. appendix D of [19], eq D10). There was footnote 3 that attempted to explain this, we have clarified it.
2- Justification have been added, all based on standard hydrodynamics as explained above. As mention, we believe explicit examples, with numerical checks, are beyond the scope of this paper; these will be provided in future works.
3- This is a good point, thank you. Indeed inhomogeneities sometimes preclude thermalization (as in MBL). But if the spatial variation of the potential is very smooth, then we expect local thermalization (as locally the systems looks homogeneous), and thus no localization. In particular, this is very different from the case of random fields. Hence, and, at large times, we expect full thermalization. Of course, hydrodynamics dramatically breaks down in the presence of localization, as the basic hydrodynamic principles are broken (local entropy maximization). But we do not expect this to occur for slowly-varying potentials. We have added few sentences (see p 7, right), and we have in fact re-organized this part of the section.
4- This was not so clear indeed; we only used the assumption that the potentials V_k(x) are varying in a smooth enough way so that we can neglect higher derivative terms. We have added a sentence "Thus, as long as ...". The point made after this sentence is that, the local GGE approximation leads to one-derivative terms only (as is clear from the conservation equations), and thus it is consistent with the approximation of neglecting two- and more-derivative terms.
5- We do not know what the solution might be - it seems to be standard practice to refer to equations in this manner.
6-Corrected
Report 91
Concerning clarifying assumptions for the validity of equations in force field.
Yes, to get eq (45) (prev (51)) indeed integrability is not ``used", and so, it is true that if there was no integrability, and only a finite number of conserved currents, then the equation would still hold, for these currents. But integrability is implied in (51), as this is an equation for all currents of the integrable theory (all currents of the local dynamics as explained above). We have added a sentence on p 7, left to that effect.
1) As explained above, equation holds under hydrodynamic approximation (local state are GGEs, higher derivatives are neglected). This is an approximation, and never becomes exact; but as in usual hydrodynamics, can be valid for a large time if variations are on long enough distances. For instance, standard Euler hydrodynamics describes quite well water waves, even if these solutions are not functions of x/t taken in the exact scaled limit (the modifications by Navier-Stokes viscosity terms make the waves loose their strength with time). This is in this hydro sense that our equations hold; up to higher-order derivatives.
Actually the sentence pointed out by the referee was not very clear. If, as time evolves, variations stay on large scales, for instance on scales of force field variations, then we can always approximate, to a good accuracy, the system by the local-GGE fluid state, at any time. The system will always, to a good accuracy, satisfy (45), and (47,48). What is expected is that solutions to our equations will go away, at large times, from real-system behaviour, because of higher-derivative terms. We would expect that these higher derivative terms will tend to make currents decay to zero. This is true of both in (45) and in (47,48).
2) We disagree. We show in appendix that (47) follows from (45), when (45) is written for all conserved current of the local dynamics (as it is in the manuscript). There is no additional assumption.
We think there is a bit of confusion here in the statement of the referee. First eq (45) is never exact: it is Euler hydrodynamics, hence cannot be exact. There is always the hydro approximation. With this choice of potential, clearly there won't be higher-derivative potential terms, but there will be higher-derivative hydrodynamic variable terms (such as viscosity). This is never an exact equation. Second, eq (47) is also not exact, but only for these reasons, not because integrability is broken. Locally integrability is not broken and things are homogeneous in the hydro approximation, so it makes sense to talk about the objects of TBA (which are only a device to write local densities and currents) and to write (47). In fact, equations like (47) without force term, that is under completely integrable dynamics as in [19,20], still are not exact - one still neglects viscosity etc. It's the same non-exactness.
There is no local density approximation made. A local density approximation is the approximation that the state at $x$ in the density matrix $e^{-\int dx \beta(x) h(x)}$ is determined as $e^{-\beta(x) H}$ (here the example is just with the hamiltonian). This is similar, but different from, the hydro approximation. In the hydro approximation, we do not say that the space-dependent potentials are determined by the local potentials of the state; we just say they satisfy the hydro equations. LDA is considered in p 7, right, in the context of the stationary solution.
We repeat that (47) is derived in appendix from (45). The only thing used are TBA expressions of local GGE averages. Nothing else. If the referee can point out another approximation in the derivation, we would be glad to see it.
Specific points:
1- Yes we agree, a review has been cited. We understand that a lot of work has been done on clarifying the role of quasi-local charges in GGE, but we believe that how the Hilbert space of Prosen's pseudolocal charges is involved in the GGE was only unambiguously established in [34], and the fact that homogeneous states thermalize to GGE proven rigorously, with the role of completeness clearly related to generalized thermalization. The latter is particularly relevant to generalized hydrodynamics.
2- We repeat the answer to point 1 of other referee: No the state is completely general, it does not need to be stationary (or homogeneous). This is because the operator $q$ itself is a linear functional of $h$, hence every matrix element is. Indeed, let $h_i$ be a basis. Then $Q[\sum_i a_i h_i] = \sum_i a_i Q[h_i]$ by linearity. Since $Qh_i = \int dx qh_i$, by locality we must also have $q[\sum_i a_i h_i] = \sum_i a_i q[h_i]$ (this is up to a total derivative of a local field, which can be set to zero by our choice of the density $qh_i$). Thus linearity holds at the operator level, and thus at the operator level $q(x) = \int d\theta h(\theta) \hat q(x,\theta)$. For the current, this then follows from the general relation between matrix elements of currents and densities (see e.g. appendix D of [19], eq D10). There was footnote 3 that attempted to explain this, we have clarified it.
3- Corrected
4- We have tried to clarify it, splitting point (i) into three sentences.
5- Corrected
6- Yes this is at the first order in a derivative expansion; comments added just above eq (37).
7- We agree that this section is a bit more difficult to read, and certainly not rigorous. Since it is not the main point of the paper, and seems to disrupt the flow of the ideas a little bit, we decided to report the whole free-particle section to appendix A, where there is no mention of "proof" in the title. We believe it is made clear enough that this is a sketch of a proof.
8- The assumption is made only in order to get conditions on the coefficient $bh$ -- it is not an assumption about the form of the state, just a technical trick to get a condition on that coefficient (as expressions (A7) are valid for general states but the coefficients do not depend on the state). We have clarified the sentence.
9- Thank you, indeed this was quite unclear. We have clarified: what we mean is that we are first proving the general equation (46) (equivalent ot (45)). We have also clarified the argument a little bit, by adding the paragraph on p 13, right, about how to get from the operator equation to the GGE-average equation (which was missing in the previous version).
10- Corrected
11- Corrected
Author: Benjamin Doyon on 2017-04-05 [id 113]
(in reply to Report 2 on 2017-03-03)We are very grateful to both referees for their careful reading of the manuscript and for their suggestions; this helped us a lot to improve the paper.
Main changes:
1) We have reported all of what was previously section IV to appendix A. That is, all discussion of emergence of GGE in free models is in the appendix. We believe this section was not as clear, and that it disrupted a little bit the flow of ideas in the paper. Since it is also not entirely rigorous, we thought it is better to put it in appendix.
2) We have added explanations concerning our assumptions in section V (force fields).
For both referees: Concerning force field (current version Sect V).
The only approximation is the hydrodynamic approximation. This is, in presence of force field, (1) the approximation that local densities and currents are in entropy-maximized states wrt local evolution, and (2) the ``smoothness" approximation which take all $x,t$-dependent quantities, including hydrodynamic fields and force fields, at first derivative order only. It is usually expected, in hydro, that correction terms to the former approximation only lead to second and higher derivatives, so the two approximations are expected to be consistent. That is, it is expected that if things are smooth enough, then locally the state is very near to a homogeneous stationary state wrt the local evolution. The first approximation involves integrability, simply because homogeneous stationary states wrt evolutions by $H+W$ are GGEs. The rest is then purely a derivation, there is no further approximation.
We understand that the referees worry about integrability breaking - clearly $H+\int dx W(x)$ is not integrable. However this does not affect the hydrodynamic description, as this description only relies on the local form of time evolution, which is integrable. An example in ordinary hydro: we know well that in ordinary hydro, a fluid cell is a galilean boost of a thermal state (e.g. it has a local density and a local momentum). This is, for the local state, a homogeneous density matrix $e^{-\beta(H-\nu P)}$ where $P$ is momentum. Now this is true also within force fields such as water in gravitational field. But if the gravitational field is space dependent, it clearly breaks translation invariance. Thus globally only the inhomogeneous hamiltonian is conserved, no momentum. Nevertheless, the fluid cell is still described in the same way, involving $P$. This is because locally, it looks homogeneous, so we recover this conserved quantity. The momentum-breaking nature of the force field appears in the fluid equations themselves, where the force term modifies the macroscopic momentum-conservation equation. It is exactly the same phenomenon here, only extended to all conserved charges. So we do not assume that integrability breaking effects are subleading wrt the terms we identify. All integrability breaking effects, at first-derivative order, are included.
We note additionally: Integrability breaking will be important in time evolution because the higher-derivative terms from force field always play a role - eventually they will make the solution go away from GHD, produce entropy, etc. Thus we would expect, at large times, the system to thermalize according to the (non-integrable) inhomogeneous hamiltonian. But we do not know the time scale for this, and there may be pre-thermalization effects.
We have added two paragraphs on p5, right, which hopefully clarifies our approximations and some of the questions the referees had.
Finally for both reports, concerning examples:
This is of course a very good comment, but unfortunately, in this area of research, nontrivial examples are rather hard to work out fully. This is because hydrodynamics describes very complicated systems in a simple way (this is its strength), so verifying it requires complicated quantum calculations. We have work in progress in this direction, but we believe it is beyond the scope of this paper. As for providing examples with a free theory, again this would be interesting, but we think it's better to study all such examples in a future work dedicated to this.
The point of this paper was to start with the hydrodynamic assumptions only, and get all logical conclusions from this. The only approximations used are those of hydrodynamics, and they have all been stated, we hope clearly in the latest version. The rest is pure derivation from first principles. Since there is no other assumption being made in deriving the main formulae, we believe this is sufficient. Of course, the hydro assumptions, and their consequences, must be checked (and have been checked in the past in ordinary hydro!). But we believe this is for future works.
Report 92:
Concerning the proof in the free-particle case (current version App A).
Hydrodynamics is only supposed to be valid, in general, in smooth enough regions. This is a strong assumption, but in general, Euler-scale hydrodynamics is not expected if this assumption is not satisfied, and so we believe our assumptions are fine in order to study emergence of hydrodynamics.
The initial domain-wall condition is of course not smooth, and hydrodynamics is not expected to correctly describes time evolution from this initial condition. This is known, and in fact has been observed numerically in [20] by comparing with DMRG. However, in [19,20], hydrodynamic is not used from initial time, it is only required at large (in fact infinite) times. Then, the profiles are very smooth, and hydrodynamic holds. In [19,20], the exact solution only requires asymptotics in space, so the precise initial condition is not necessary. In the present manuscript, we show that, if profiles are smooth enough, indeed one can use hydrodynamics, and thus this provides a missing step in [19,20] in the simple case of free models. Of course, free models can be solved by other means, the result here is of conceptual rather than practical interest.
Concerning the conservation laws (Sect V).
It is true that a particular conservation law can have many more commuting charges. However, the main precept of hydrodynamics is that, locally, the system maximize entropy with respect to all commuting charges conserved by the local dynamics. Here, the local dynamics is written in the form $H+W$, where $W$ involves higher conserved charges. So, we must look for the commuting charges with respect to $H+W$. If we take $W = Q_0-H$, then we agree that locally there would be many more conservation laws (the local dynamics, wrt $Q_0$, being trivial). But with $W$ a generic linear combination of higher-spin conserved charges, this is not the case: they have the same set of commuting operators as $H$ itself. This is the basic structure of integrability, any higher-spin conserved quantity can be considered as a hamiltonian, and it has the same set of commuting conserved quantities (same integrability class). This is what we assume: genericity of $W$. In this case, the quasi-particle description is expected to be sufficient (cf [19,20]). Indeed this was not explicitly mentioned, we have added two sentences, page 4, right.
Specific points:
1- No the state is completely general, it does not need to be stationary (or homogeneous). This is because the operator $q$ itself is a linear functional of $h$, hence every matrix element is. Indeed, let $h_i$ be a basis. Then $Q[\sum_i a_i h_i] = \sum_i a_i Q[h_i]$ by linearity. Since $Qh_i = \int dx qh_i$, by locality we must also have $q[\sum_i a_i h_i] = \sum_i a_i q[h_i]$ (this is up to a total derivative of a local field, which can be set to zero by our choice of the density $qh_i$). Thus linearity holds at the operator level, and thus $q(x) = \int d\theta h(\theta) \hat q(x,\theta)$. For the current, this then follows from the general relation between matrix elements of currents and densities (see e.g. appendix D of [19], eq D10). There was footnote 3 that attempted to explain this, we have clarified it.
2- Justification have been added, all based on standard hydrodynamics as explained above. As mention, we believe explicit examples, with numerical checks, are beyond the scope of this paper; these will be provided in future works.
3- This is a good point, thank you. Indeed inhomogeneities sometimes preclude thermalization (as in MBL). But if the spatial variation of the potential is very smooth, then we expect local thermalization (as locally the systems looks homogeneous), and thus no localization. In particular, this is very different from the case of random fields. Hence, and, at large times, we expect full thermalization. Of course, hydrodynamics dramatically breaks down in the presence of localization, as the basic hydrodynamic principles are broken (local entropy maximization). But we do not expect this to occur for slowly-varying potentials. We have added few sentences (see p 7, right), and we have in fact re-organized this part of the section.
4- This was not so clear indeed; we only used the assumption that the potentials V_k(x) are varying in a smooth enough way so that we can neglect higher derivative terms. We have added a sentence "Thus, as long as ...". The point made after this sentence is that, the local GGE approximation leads to one-derivative terms only (as is clear from the conservation equations), and thus it is consistent with the approximation of neglecting two- and more-derivative terms.
5- We do not know what the solution might be - it seems to be standard practice to refer to equations in this manner.
6-Corrected
Report 91
Concerning clarifying assumptions for the validity of equations in force field.
Yes, to get eq (45) (prev (51)) indeed integrability is not ``used", and so, it is true that if there was no integrability, and only a finite number of conserved currents, then the equation would still hold, for these currents. But integrability is implied in (51), as this is an equation for all currents of the integrable theory (all currents of the local dynamics as explained above). We have added a sentence on p 7, left to that effect.
1) As explained above, equation holds under hydrodynamic approximation (local state are GGEs, higher derivatives are neglected). This is an approximation, and never becomes exact; but as in usual hydrodynamics, can be valid for a large time if variations are on long enough distances. For instance, standard Euler hydrodynamics describes quite well water waves, even if these solutions are not functions of x/t taken in the exact scaled limit (the modifications by Navier-Stokes viscosity terms make the waves loose their strength with time). This is in this hydro sense that our equations hold; up to higher-order derivatives.
Actually the sentence pointed out by the referee was not very clear. If, as time evolves, variations stay on large scales, for instance on scales of force field variations, then we can always approximate, to a good accuracy, the system by the local-GGE fluid state, at any time. The system will always, to a good accuracy, satisfy (45), and (47,48). What is expected is that solutions to our equations will go away, at large times, from real-system behaviour, because of higher-derivative terms. We would expect that these higher derivative terms will tend to make currents decay to zero. This is true of both in (45) and in (47,48).
2) We disagree. We show in appendix that (47) follows from (45), when (45) is written for all conserved current of the local dynamics (as it is in the manuscript). There is no additional assumption.
We think there is a bit of confusion here in the statement of the referee. First eq (45) is never exact: it is Euler hydrodynamics, hence cannot be exact. There is always the hydro approximation. With this choice of potential, clearly there won't be higher-derivative potential terms, but there will be higher-derivative hydrodynamic variable terms (such as viscosity). This is never an exact equation. Second, eq (47) is also not exact, but only for these reasons, not because integrability is broken. Locally integrability is not broken and things are homogeneous in the hydro approximation, so it makes sense to talk about the objects of TBA (which are only a device to write local densities and currents) and to write (47). In fact, equations like (47) without force term, that is under completely integrable dynamics as in [19,20], still are not exact - one still neglects viscosity etc. It's the same non-exactness.
There is no local density approximation made. A local density approximation is the approximation that the state at $x$ in the density matrix $e^{-\int dx \beta(x) h(x)}$ is determined as $e^{-\beta(x) H}$ (here the example is just with the hamiltonian). This is similar, but different from, the hydro approximation. In the hydro approximation, we do not say that the space-dependent potentials are determined by the local potentials of the state; we just say they satisfy the hydro equations. LDA is considered in p 7, right, in the context of the stationary solution.
We repeat that (47) is derived in appendix from (45). The only thing used are TBA expressions of local GGE averages. Nothing else. If the referee can point out another approximation in the derivation, we would be glad to see it.
Specific points:
1- Yes we agree, a review has been cited. We understand that a lot of work has been done on clarifying the role of quasi-local charges in GGE, but we believe that how the Hilbert space of Prosen's pseudolocal charges is involved in the GGE was only unambiguously established in [34], and the fact that homogeneous states thermalize to GGE proven rigorously, with the role of completeness clearly related to generalized thermalization. The latter is particularly relevant to generalized hydrodynamics.
2- We repeat the answer to point 1 of other referee: No the state is completely general, it does not need to be stationary (or homogeneous). This is because the operator $q$ itself is a linear functional of $h$, hence every matrix element is. Indeed, let $h_i$ be a basis. Then $Q[\sum_i a_i h_i] = \sum_i a_i Q[h_i]$ by linearity. Since $Qh_i = \int dx qh_i$, by locality we must also have $q[\sum_i a_i h_i] = \sum_i a_i q[h_i]$ (this is up to a total derivative of a local field, which can be set to zero by our choice of the density $qh_i$). Thus linearity holds at the operator level, and thus at the operator level $q(x) = \int d\theta h(\theta) \hat q(x,\theta)$. For the current, this then follows from the general relation between matrix elements of currents and densities (see e.g. appendix D of [19], eq D10). There was footnote 3 that attempted to explain this, we have clarified it.
3- Corrected
4- We have tried to clarify it, splitting point (i) into three sentences.
5- Corrected
6- Yes this is at the first order in a derivative expansion; comments added just above eq (37).
7- We agree that this section is a bit more difficult to read, and certainly not rigorous. Since it is not the main point of the paper, and seems to disrupt the flow of the ideas a little bit, we decided to report the whole free-particle section to appendix A, where there is no mention of "proof" in the title. We believe it is made clear enough that this is a sketch of a proof.
8- The assumption is made only in order to get conditions on the coefficient $bh$ -- it is not an assumption about the form of the state, just a technical trick to get a condition on that coefficient (as expressions (A7) are valid for general states but the coefficients do not depend on the state). We have clarified the sentence.
9- Thank you, indeed this was quite unclear. We have clarified: what we mean is that we are first proving the general equation (46) (equivalent ot (45)). We have also clarified the argument a little bit, by adding the paragraph on p 13, right, about how to get from the operator equation to the GGE-average equation (which was missing in the previous version).
10- Corrected
11- Corrected

---

## Round 2 · Author Response

We are very grateful to both referees for their careful reading of the manuscript and for their suggestions; this helped us a lot to improve the paper.
Main changes:
1) We have reported all of what was previously section IV to appendix A. That is, all discussion of emergence of GGE in free models is in the appendix. We believe this section was not as clear, and that it disrupted a little bit the flow of ideas in the paper. Since it is also not entirely rigorous, we thought it is better to put it in appendix.
2) We have added explanations concerning our assumptions in section V (force fields).
For both referees: Concerning force field (current version Sect V).
The only approximation is the hydrodynamic approximation. This is, in presence of force field, (1) the approximation that local densities and currents are in entropy-maximized states wrt local evolution, and (2) the ``smoothness" approximation which take all $x,t$-dependent quantities, including hydrodynamic fields and force fields, at first derivative order only. It is usually expected, in hydro, that correction terms to the former approximation only lead to second and higher derivatives, so the two approximations are expected to be consistent. That is, it is expected that if things are smooth enough, then locally the state is very near to a homogeneous stationary state wrt the local evolution. The first approximation involves integrability, simply because homogeneous stationary states wrt evolutions by $H+W$ are GGEs. The rest is then purely a derivation, there is no further approximation.
We understand that the referees worry about integrability breaking - clearly $H+\int dx W(x)$ is not integrable. However this does not affect the hydrodynamic description, as this description only relies on the local form of time evolution, which is integrable. An example in ordinary hydro: we know well that in ordinary hydro, a fluid cell is a galilean boost of a thermal state (e.g. it has a local density and a local momentum). This is, for the local state, a homogeneous density matrix $e^{-\beta(H-\nu P)}$ where $P$ is momentum. Now this is true also within force fields such as water in gravitational field. But if the gravitational field is space dependent, it clearly breaks translation invariance. Thus globally only the inhomogeneous hamiltonian is conserved, no momentum. Nevertheless, the fluid cell is still described in the same way, involving $P$. This is because locally, it looks homogeneous, so we recover this conserved quantity. The momentum-breaking nature of the force field appears in the fluid equations themselves, where the force term modifies the macroscopic momentum-conservation equation. It is exactly the same phenomenon here, only extended to all conserved charges. So we do not assume that integrability breaking effects are subleading wrt the terms we identify. All integrability breaking effects, at first-derivative order, are included.
We note additionally: Integrability breaking will be important in time evolution because the higher-derivative terms from force field always play a role - eventually they will make the solution go away from GHD, produce entropy, etc. Thus we would expect, at large times, the system to thermalize according to the (non-integrable) inhomogeneous hamiltonian. But we do not know the time scale for this, and there may be pre-thermalization effects.
We have added two paragraphs on p5, right, which hopefully clarifies our approximations and some of the questions the referees had.
Finally for both reports, concerning examples:
This is of course a very good comment, but unfortunately, in this area of research, nontrivial examples are rather hard to work out fully. This is because hydrodynamics describes very complicated systems in a simple way (this is its strength), so verifying it requires complicated quantum calculations. We have work in progress in this direction, but we believe it is beyond the scope of this paper. As for providing examples with a free theory, again this would be interesting, but we think it's better to study all such examples in a future work dedicated to this.
The point of this paper was to start with the hydrodynamic assumptions only, and get all logical conclusions from this. The only approximations used are those of hydrodynamics, and they have all been stated, we hope clearly in the latest version. The rest is pure derivation from first principles. Since there is no other assumption being made in deriving the main formulae, we believe this is sufficient. Of course, the hydro assumptions, and their consequences, must be checked (and have been checked in the past in ordinary hydro!). But we believe this is for future works.
Report 92:
Concerning the proof in the free-particle case (current version App A).
Hydrodynamics is only supposed to be valid, in general, in smooth enough regions. This is a strong assumption, but in general, Euler-scale hydrodynamics is not expected if this assumption is not satisfied, and so we believe our assumptions are fine in order to study emergence of hydrodynamics.
The initial domain-wall condition is of course not smooth, and hydrodynamics is not expected to correctly describes time evolution from this initial condition. This is known, and in fact has been observed numerically in [20] by comparing with DMRG. However, in [19,20], hydrodynamic is not used from initial time, it is only required at large (in fact infinite) times. Then, the profiles are very smooth, and hydrodynamic holds. In [19,20], the exact solution only requires asymptotics in space, so the precise initial condition is not necessary. In the present manuscript, we show that, if profiles are smooth enough, indeed one can use hydrodynamics, and thus this provides a missing step in [19,20] in the simple case of free models. Of course, free models can be solved by other means, the result here is of conceptual rather than practical interest.
Concerning the conservation laws (Sect V).
It is true that a particular conservation law can have many more commuting charges. However, the main precept of hydrodynamics is that, locally, the system maximize entropy with respect to all commuting charges conserved by the local dynamics. Here, the local dynamics is written in the form $H+W$, where $W$ involves higher conserved charges. So, we must look for the commuting charges with respect to $H+W$. If we take $W = Q_0-H$, then we agree that locally there would be many more conservation laws (the local dynamics, wrt $Q_0$, being trivial). But with $W$ a generic linear combination of higher-spin conserved charges, this is not the case: they have the same set of commuting operators as $H$ itself. This is the basic structure of integrability, any higher-spin conserved quantity can be considered as a hamiltonian, and it has the same set of commuting conserved quantities (same integrability class). This is what we assume: genericity of $W$. In this case, the quasi-particle description is expected to be sufficient (cf [19,20]). Indeed this was not explicitly mentioned, we have added two sentences, page 4, right.
Specific points:
1- No the state is completely general, it does not need to be stationary (or homogeneous). This is because the operator $q$ itself is a linear functional of $h$, hence every matrix element is. Indeed, let $h_i$ be a basis. Then $Q[\sum_i a_i h_i] = \sum_i a_i Q[h_i]$ by linearity. Since $Qh_i = \int dx qh_i$, by locality we must also have $q[\sum_i a_i h_i] = \sum_i a_i q[h_i]$ (this is up to a total derivative of a local field, which can be set to zero by our choice of the density $qh_i$). Thus linearity holds at the operator level, and thus $q(x) = \int d\theta h(\theta) \hat q(x,\theta)$. For the current, this then follows from the general relation between matrix elements of currents and densities (see e.g. appendix D of [19], eq D10). There was footnote 3 that attempted to explain this, we have clarified it.
2- Justification have been added, all based on standard hydrodynamics as explained above. As mention, we believe explicit examples, with numerical checks, are beyond the scope of this paper; these will be provided in future works.
3- This is a good point, thank you. Indeed inhomogeneities sometimes preclude thermalization (as in MBL). But if the spatial variation of the potential is very smooth, then we expect local thermalization (as locally the systems looks homogeneous), and thus no localization. In particular, this is very different from the case of random fields. Hence, and, at large times, we expect full thermalization. Of course, hydrodynamics dramatically breaks down in the presence of localization, as the basic hydrodynamic principles are broken (local entropy maximization). But we do not expect this to occur for slowly-varying potentials. We have added few sentences (see p 7, right), and we have in fact re-organized this part of the section.
4- This was not so clear indeed; we only used the assumption that the potentials V_k(x) are varying in a smooth enough way so that we can neglect higher derivative terms. We have added a sentence "Thus, as long as ...". The point made after this sentence is that, the local GGE approximation leads to one-derivative terms only (as is clear from the conservation equations), and thus it is consistent with the approximation of neglecting two- and more-derivative terms.
5- We do not know what the solution might be - it seems to be standard practice to refer to equations in this manner.
6-Corrected
Report 91
Concerning clarifying assumptions for the validity of equations in force field.
Yes, to get eq (45) (prev (51)) indeed integrability is not ``used", and so, it is true that if there was no integrability, and only a finite number of conserved currents, then the equation would still hold, for these currents. But integrability is implied in (51), as this is an equation for all currents of the integrable theory (all currents of the local dynamics as explained above). We have added a sentence on p 7, left to that effect.
1) As explained above, equation holds under hydrodynamic approximation (local state are GGEs, higher derivatives are neglected). This is an approximation, and never becomes exact; but as in usual hydrodynamics, can be valid for a large time if variations are on long enough distances. For instance, standard Euler hydrodynamics describes quite well water waves, even if these solutions are not functions of x/t taken in the exact scaled limit (the modifications by Navier-Stokes viscosity terms make the waves loose their strength with time). This is in this hydro sense that our equations hold; up to higher-order derivatives.
Actually the sentence pointed out by the referee was not very clear. If, as time evolves, variations stay on large scales, for instance on scales of force field variations, then we can always approximate, to a good accuracy, the system by the local-GGE fluid state, at any time. The system will always, to a good accuracy, satisfy (45), and (47,48). What is expected is that solutions to our equations will go away, at large times, from real-system behaviour, because of higher-derivative terms. We would expect that these higher derivative terms will tend to make currents decay to zero. This is true of both in (45) and in (47,48).
2) We disagree. We show in appendix that (47) follows from (45), when (45) is written for all conserved current of the local dynamics (as it is in the manuscript). There is no additional assumption.
We think there is a bit of confusion here in the statement of the referee. First eq (45) is never exact: it is Euler hydrodynamics, hence cannot be exact. There is always the hydro approximation. With this choice of potential, clearly there won't be higher-derivative potential terms, but there will be higher-derivative hydrodynamic variable terms (such as viscosity). This is never an exact equation. Second, eq (47) is also not exact, but only for these reasons, not because integrability is broken. Locally integrability is not broken and things are homogeneous in the hydro approximation, so it makes sense to talk about the objects of TBA (which are only a device to write local densities and currents) and to write (47). In fact, equations like (47) without force term, that is under completely integrable dynamics as in [19,20], still are not exact - one still neglects viscosity etc. It's the same non-exactness.
There is no local density approximation made. A local density approximation is the approximation that the state at $x$ in the density matrix $e^{-\int dx \beta(x) h(x)}$ is determined as $e^{-\beta(x) H}$ (here the example is just with the hamiltonian). This is similar, but different from, the hydro approximation. In the hydro approximation, we do not say that the space-dependent potentials are determined by the local potentials of the state; we just say they satisfy the hydro equations. LDA is considered in p 7, right, in the context of the stationary solution.
We repeat that (47) is derived in appendix from (45). The only thing used are TBA expressions of local GGE averages. Nothing else. If the referee can point out another approximation in the derivation, we would be glad to see it.
Specific points:
1- Yes we agree, a review has been cited. We understand that a lot of work has been done on clarifying the role of quasi-local charges in GGE, but we believe that how the Hilbert space of Prosen's pseudolocal charges is involved in the GGE was only unambiguously established in [34], and the fact that homogeneous states thermalize to GGE proven rigorously, with the role of completeness clearly related to generalized thermalization. The latter is particularly relevant to generalized hydrodynamics.
2- We repeat the answer to point 1 of other referee: No the state is completely general, it does not need to be stationary (or homogeneous). This is because the operator $q$ itself is a linear functional of $h$, hence every matrix element is. Indeed, let $h_i$ be a basis. Then $Q[\sum_i a_i h_i] = \sum_i a_i Q[h_i]$ by linearity. Since $Qh_i = \int dx qh_i$, by locality we must also have $q[\sum_i a_i h_i] = \sum_i a_i q[h_i]$ (this is up to a total derivative of a local field, which can be set to zero by our choice of the density $qh_i$). Thus linearity holds at the operator level, and thus at the operator level $q(x) = \int d\theta h(\theta) \hat q(x,\theta)$. For the current, this then follows from the general relation between matrix elements of currents and densities (see e.g. appendix D of [19], eq D10). There was footnote 3 that attempted to explain this, we have clarified it.
3- Corrected
4- We have tried to clarify it, splitting point (i) into three sentences.
5- Corrected
6- Yes this is at the first order in a derivative expansion; comments added just above eq (37).
7- We agree that this section is a bit more difficult to read, and certainly not rigorous. Since it is not the main point of the paper, and seems to disrupt the flow of the ideas a little bit, we decided to report the whole free-particle section to appendix A, where there is no mention of "proof" in the title. We believe it is made clear enough that this is a sketch of a proof.
8- The assumption is made only in order to get conditions on the coefficient $bh$ -- it is not an assumption about the form of the state, just a technical trick to get a condition on that coefficient (as expressions (A7) are valid for general states but the coefficients do not depend on the state). We have clarified the sentence.
9- Thank you, indeed this was quite unclear. We have clarified: what we mean is that we are first proving the general equation (46) (equivalent ot (45)). We have also clarified the argument a little bit, by adding the paragraph on p 13, right, about how to get from the operator equation to the GGE-average equation (which was missing in the previous version).
10- Corrected
11- Corrected

---

## Round 2 · List of Changes

1) We have reported all of what was previously section IV to appendix A. That is, all discussion of emergence of GGE in free models is in the appendix. We believe this section was not as clear, and that it disrupted a little bit the flow of ideas in the paper. Since it is also not entirely rigorous, we thought it is better to put it in appendix.
2) We have added explanations concerning our assumptions in section V (force fields).
3) we have made the small modifications (typos, etc) suggested by the referees

---

## Editorial Decision

published